



# Weathering without inorganic CDR revealed through cation tracing.

Arthur Vienne[1] , Patrick Frings[2] , Jet Rijnders[1] , Tim Jesper suhrhoff[4,5], Tom Reershemius[6], Reinaldy P. Poetra[3] , Jens Hartmann[3] , Harun Niron[1], Miguel Portillo Estrada[1],Laura Steinwidder[1], Lucilla Boito[1], Sara Vicca[1]

1Biobased Sustainability Engineering (SUSTAIN), Department of Bioscience Engineering, University of Antwerp, Antwerp, Belgium
GFZ German Research Centre for Geosciences, Section Earth Surface Geochemistry, Telegrafenberg, 14473 Potsdam, Germany
Institute for Geology, Centre for Earth System Research and Sustainability (CEN), Universität Hamburg, Bundesstraße 55, 20146 Hamburg, Germany
Yale Center for Natural Carbon Capture, Yale University, New Haven, CT 06511, USA
Department of Earth and Planetary Sciences, Yale University, New Haven, CT 06511, USA
School of Natural and Environmental Sciences, Newcastle University, Newcastle upon Tyne, UK

Correspondance to: arthur.vienne@uantwerpen.be ; sara.vicca@uantwerpen.be

Keywords: CDR, Enhanced weathering, MRV, sequential extractions, weathering, basalt, time lags for CDR, secondary minerals

Graphical abstract

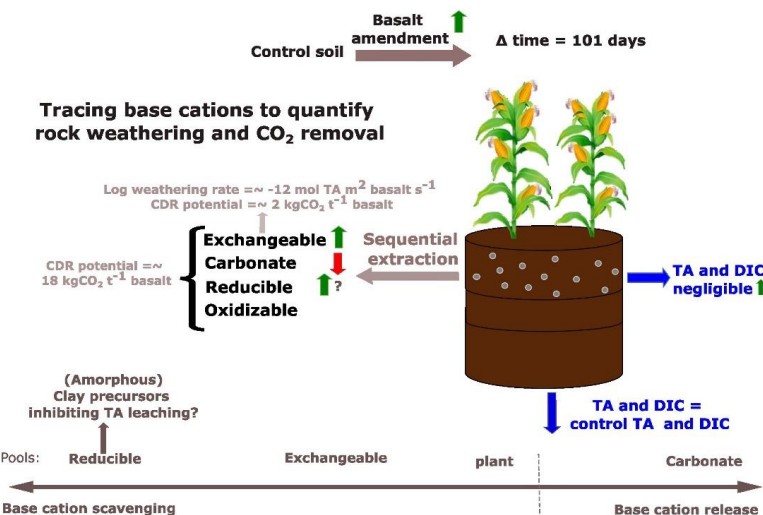

# Abstract

Enhanced Weathering using basalt rock dust is a scalable carbon dioxide removal (CDR) technique, but quantifying rock weathering and CDR rates poses a critical challenge. Here, we investigated inorganic CDR and weathering rates by treating mesocosms planted with corn with basalt (0, 10, 30, 50, 75, 100, 150 and  200 t ha$^{-1}$) and monitoring them for 101 days. Surprisingly, we observed no significant inorganic CDR, as leaching of dissolved inorganic carbon did not increase, and soil carbonate content even declined over time.

To gain insights into the weathering processes, we analyzed the mass balance of base cations, which can be linked with anions (including $HCO_3^-$) through charge balance. This mass balance showed that most base cation charges were retained as (hydr)oxides in the reducible pool of the top soil, while increases in the exchangeable pool were about a factor 10 smaller. Soil base cation scavenging exceeded plant scavenging by approximately two orders of magnitude. From the base cations in all pools (soil, soil water and plants), we quantified log weathering rates of -11 mol TA m$^{-2}$ basalt s$^{-1}$ and a maximum $CO_2$ removal potential of the weathered base cations (i.e., CDR potential) of 18 kg $CO_2$ t$^{-1}$ basalt.



For climate change mitigation, not only the amount of CDR potential is important, but also the timescale at which
that CDR would be realized. Our data suggests that the lag time for realization of inorganic CDR may be larger than
commonly assumed. In conclusion, we observed that inorganic CDR was not directly linked to rock weathering in
the short-term. Still, the observed increases in secondary minerals and base cation exchange may provide valuable
benefits for soil fertility and organic matter stabilization in the long-term.

# 1. Introduction

To meet the "well below 2°C warming" target established by the United Nations' Paris Agreement, Carbon Dioxide
Removal (CDR) must complement conventional climate change mitigation efforts (Minx et al., 2018). One CDR
technology under consideration is enhanced weathering (EW). EW relies on accelerating natural weathering
reactions of silicate minerals with $H_2O$ and $CO_2$ (as in **Reactions 1 to 3**), which increases the concentration of base
cations and dissolved inorganic C (DIC) in water. As a proxy for DIC, total alkalinity (TA) is often used, which can
be approximated as the sum of base cation charges (**Equation 1**)(Amann & Hartmann, 2022; Barker, 2013; Wolf-
Gladow et al., 2007).

$$Mg_2SiO_4 + 4CO_2 + 4 H_2O \rightarrow 2\ Mg^{2+} + 4HCO_3^- + H_4SiO_4$$ **Reaction 1**
(Mg-olivine weathering)

$$Ca_{0.5}Na_{0.5}Al_{1.5}Si_{2.5}O_8 + 1.5\ CO_2 + 8\ H_2O \rightarrow 0.5\ Na^+ + 0.5\ Ca^{2+} + 1.5\ HCO_3^- + 1.5\ Al(OH)_3 + 2.5\ H_4SiO_4$$ **Reaction 2**
(plagioclase (labradorite) weathering)

$$MgCaSi_2O_6 + 4CO_2 + 6\ H_2O \rightarrow Ca^{2+} + Mg^{2+} + 4HCO_3^- + 2\ H_4SiO_4$$ **Reaction 3**
(pyroxene (diopside) weathering)

$$\Delta TA \approx 2 * (\Delta Ca + \Delta Mg) + \Delta Na + \Delta K \qquad (1)$$

EW is an attractive CDR technology for several reasons. First, EW may provide long-lived to permanent $CO_2$
sequestration: if fixed, DIC is transported via rivers or groundwater to oceans where it may not be released back
into the atmosphere for millennia, the timescale needed for oceanic carbonate precipitation, which would release
50% of the DIC input back into the atmosphere (**Reaction 4**) (Renforth & Henderson, 2017). Secondly, rock dust
amendment has the potential to improve soil fertility and counters soil acidification (Swoboda et al., 2021; Van
Straaten, 2006). Thirdly, unlike some other CDR technologies (such as bio-energy with carbon capture and storage
(BECCS) or afforestation), EW avoids competition for land with food production (Fuss et al., 2018; Janssens et al.,
2022; Smith et al., 2016). Although several rock types are considered for EW, basalt is typically used in EW field
trials and has several advantages. Basalt has relatively high base cation content, particularly of $Ca^{2+}$ and $Mg^{2+}$ ,
which translates into a high potential for $CO_2$ removal (Renforth et al., 2019). Additionally, basalt is comprised of
mafic silicate minerals such as plagioclases, pyroxene, and olivine, known for their relatively high weathering rates
(Wr). Furthermore, basalt formations are abundant, widely distributed and close to major economies, making the
adoption of EW using basalt scalable. Importantly, basalt is safer for agricultural application compared to ultramafic
rocks like dunite  due to its lower content of heavy metals such as Ni and Cr (Beerling et al., 2020).



Despite the great potential of terrestrial EW and substantial attention by industry in recent years, monitoring rock
weathering and CDR is challenging. Quantification of inorganic CDR by EW has often focussed on tracking DIC or
alkalinity leaching in porewaters and drainage (Holzer et al., 2023; Larkin et al., 2022). However, recent studies
have shown that soils can greatly influence DIC leaching dynamics (Dietzen et al., 2018; Niron et al., 2024; Reynaert
et al., 2023; Vienne et al., 2024). DIC can for example precipitate as soil inorganic carbon (SIC) in the form of solid
carbonates, thereby losing half of the initially captured $CO_2$ (**Reaction 4**)(Haque et al., 2019). A robust and reliable
accounting of inorganic CDR must thus include both monitoring of DIC leaching and SIC changes.

$$2\ HCO_3^- + Ca^{2+} \rightarrow CaCO_3 + H_2O + CO_2 \hspace{3em} \textbf{Reaction 4}$$

Focusing solely on changes in DIC and SIC may however overlook other critical soil processes that impact CDR.
Besides the carbonate soil pool, other solid soil pools can also extract base cations from solution (**Figure 1**). These
pools (temporally) trap base cations, preventing DIC leaching and could stabilize soil organic matter (SOM) (Buss
et al., 2024). Here, we trace the fate of cations in four different soil pools, to gain better estimates of Wrs and CDR.

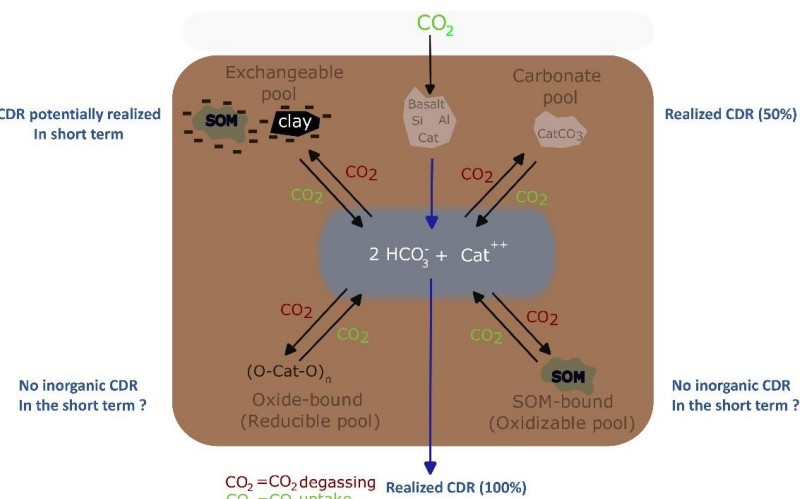


**Figure 1:** Schematized weathering of aluminosilicate rock and four soil pools that scavenge base cations (=
alkalinity): exchangeable pool, carbonate pool, reducible pool and oxidizable pool. Because of charge balance,
uptake of base cations by these pools releases H+ that can reconvert $HCO_3^-$ (that was a priori generated from $CO_2$
through weathering) into $CO_2$. Cat++ = one divalent base cation or two monovalent base cations. In each corner,
in blue, the significance for CDR is indicated for each of the soil pools.
Tracing base cations in soils can be done based on the established methodology of Tessier et al. (1979) in which
cations are partitioned into four operationally defined soil pools; The exchangeable pool, the carbonate pool, the
reducible pool and the oxidizable pool. In the exchangeable soil pool, cations interact with negatively charged clay
or SOM surfaces. The carbonate pool contains carbonates such as $CaCO_3$ and the C in this pool is reported as
SIC. The detection of changes in SIC in basalt amended soils in short-duration experiments is typically challenging
(Kelland et al., 2020; Vienne et al., 2022). Focusing on carbonate base cations may avoid typical issues with C
heterogeneity and detection of relatively small SIC changes. Tessier et al. (1979) operationally defined a reducible



soil pool (oxidized cations associated with Fe and Al hydroxides) and an oxidizable pool (cations that are bound to
SOM). Cations in the oxidizable pool are expected to chemically stabilize organic matter due to inhibition of
decomposing enzymes. Last, besides soil pools, also plants can scavenge base cations from solution. Base cations
that go to the plant pool can be recycled to the aqueous phase, either through decomposition of plants in the field
or through the food chain and sewage system, further complicating base cation mass balancing.
The undesirable side-effect of base cation scavenging (by plant/soil pools) is release of $CO_2$. By charge balance,
these pools release equivalent charges of protons in return for base cations (**Figure 1**). Protons then convert
negatively charged DIC ($HCO_3^-$ and $CO_3^{2-}$) to $H_2CO_3$ which is in equilibrium with gaseous $CO_2$ ($CO_3^{2-} + H^+ \rightarrow HCO_3^-$
and $HCO_3^- + H^+ \rightarrow H_2CO_3 \Leftrightarrow H_2O + CO_2$ (g)). Hence, Inorganic CDR is reversed during storage of base cations.
Hereafter, we refer to scavenged base cations as 'scavenged TA'. Once base cations are released from these pools
into the soil water, they resequester $CO_2$. From scavenged TA, we can thus calculate a 'CDR potential'. This is a
maximum quantity of inorganic CDR that can be achieved after base cation leaching from soil pools.
Base cation retention in different soil pools results in a temporal decoupling between weathering and inorganic
CDR. The timeframe in which the CDR potential can be achieved is a major uncertainty in EW (Kanzaki et al.,
2024a). For weakly bound exchangeable cations, CDR potential may be achieved in the relative short term of
decades. Within this timeframe, because of stronger binding strengths, reducible and oxidizable base cations are
more unlikely to be released and thus deliver inorganic CDR. Last, inorganic CDR is only achieved if the weathering
agent that induced the weathering was $H_2CO_3$ (as in Reaction 1-3). If the weathering agent is another acid (e.g.
$HNO_3$ from fertilizers), no inorganic CDR occurs (McDermott et al., 2024; Taylor et al., 2020).
In a mesocosm experiment with basalt rock powder addition, we aimed to accurately quantify the Wr and CDR
potential through quantification of base cations in the four abovementioned soil pools, soil water and maize plants.
Tracing the fate of alkalinity after its generation by the weathering of primary minerals is key to accurately quantify
basalt Wrs. Here, we make a mass balance after 101 days of experiment, investigate the fate of base cations
through exploration of sequential extractions as a monitoring, reporting and verification (MRV) strategy and the
implications for CDR.

## 2. Materials and Methods

### 2.1 experimental set-up

A mesocosm experiment with 30 mesocosms was constructed at the experimental site at the Drie Eiken Campus
of the University of Antwerp (Belgium). This experiment was part of a larger mesocosm experiment that aimed to
investigate heavy metal fate in silicate amended maize plants (Rijnders et al., 2024). The mesocosms (0.6 m height,
radius=0.25m) received natural rainfall and received additional water through manual irrigation (**Fig. S2**). In May
2021, the lower 40 cm of each mesocosm was filled with a slightly acidic sandy loam soil (**Table 1**).

117                        **Table 1:** properties of control soil.



| Control soil properties* | |
|---|---|
| **pH**<br>(in a soil: water suspension (1:2.5)) | 5.66 ± 0.01 |
| **Texture**<br>(Sand, clay, silt %) | **Sandy loam**<br>(61, 4, 35 %) |
| **SOC (%)\*\*** | 0.53 ± 0.01 |
| **SIC (%)** | 0.0031 ± 0.0002 |
| **Cation exchange capacity (CEC)**<br>**(**meq/100g) | 3.03 ±0.11 |
| **Base saturation (%)** | 50 ± 5 |
| **Bulk density (BD) (kg/L)** | 1.58±0.02 |

*Reported values represent the average ± standard error (SE) of control soil sampled at all depths after the experimental period of 101 days. ** Determined through loss on ignition (4h heating at 360°C and assuming a SOC/SOM ratio of 0.58 (Van Bemmelen, 1890)).

The upper 20 cm was filled with the same soil, either unamended in the control (C) treatment (5 mesocosms), or amended with basalt (**Figure 2**). Five mesocosms received 50 ton basalt ha-1, while six others received different amounts of basalt, ranging between 10 and 200 ton/ha (**Table 2**). The basalt was mixed homogenously in the control soil using a concrete mixer. Basalt was obtained from DURABAS (https://www.rpbl.de. Particle size distribution (PSD) was analyzed using a mastersizer 2000 with a Hydro 2000G sample dispersion unit after removing larger particles with a 2 mm sieve. The P80 was 310.78 μm (see **Fig. S5**). The SSA was determined with a Quantachrome Autosorb iQ using the Braunauer-Emmet-Teller (BET) method. The measurement used nitrogen ($N_2$) as adsorbate with multi-point (5 points) and isotherm (77K) settings. Samples with the same treatment were pooled in equal quantities into one sample to reduce the cost and time for analysis. All samples were degassed at 300 ºC with 200 minutes of soak time. High measurement quality was ensured by frequent reference (Bundesanstalt für Materialforschung und -prüfung, Germany) measurements in addition to three technical repetitions for each measurement. The BET-SSA of the basalt rock was 9.226 ± 0. 08 m² g$^{-1}$. X-ray diffraction (XRD) and x-ray fluorescence (XRF) analyses are provided in the supplement.

**Table 2**: Overview of basalt application rates. The 0 and 50 t basalt ha-1 application rates were replicated in five mesocosms, while other application rates were only tested in one mesocosm. We added these replicates within individual application rates to learn about the variability between mesocosms receiving the same treatment.

| Ton silicate/ha<br>(replications) | 0<br>(5x) | 10 | 30 | 50<br>(5x) | 75 | 100 | 150 | 200 |
|---|---|---|---|---|---|---|---|---|

Basalt was mixed into the top soil on 17/5/2021. To allow leachate collection, mesocosms had a 2 cm diameter hole at the bottom. On the inside, the bottom of the pot was covered with a root exclusion mesh to prevent soil export. Glass collectors (2.3L volume) were placed under the mesocosm to collect the leachates. Leachate volumes were determined throughout the experiment and were collected for chemical analyses on seven occasions. On 3/6/2021, two sweet corn seedlings (variety Tom Thumb) were planted in each the mesocosms and all pots received fertilization with NPK (96 – 10 – 79) kg ha$^{-1}$ by adding $Ca(NO_3)_2$, triple super phosphate (TSP, 45% $P_2O_5$) and $K_2SO_4$. The experimental duration was 101 days; plants were harvested on 26/8/2021.



Soil water content and temperature were recorded using Cambell Scientific sensors (CS616) that are 30 cm in
length. Watering (using rain water collected from a tank) was executed manually and total water manual inputs
were tracked. In addition, daily precipitation amounts (in mm) were obtained for Wilrijk (Belgium) using the open
source tool (visualcrossing.com). In the supplement, an overview of environmental conditions (rainfall, total water
inputs, temperature and soil moisture) is given.

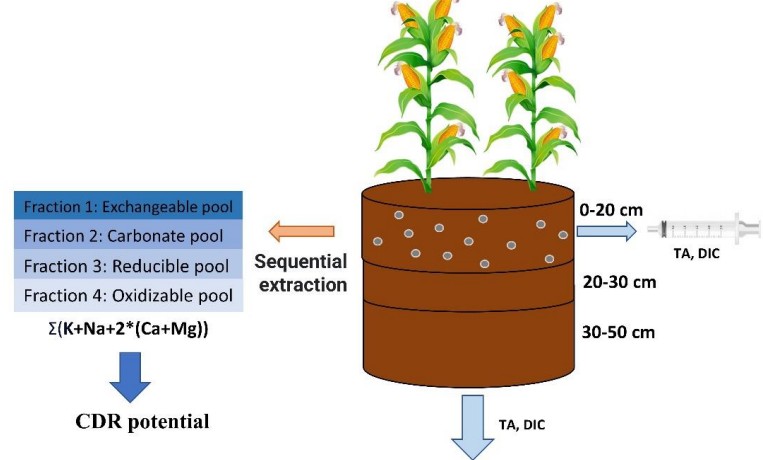


**Figure 2:** Overview of experimental set-up and measurements.
**2.2 Leachate and pore water analysis**
Weekly pore water sampling was performed with rhizons (Rhizon Flex, Rhizosphere Research Products B.V.,
Wageningen, NL) installed at 5 cm depth in each mesocosm. Leachate and porewater samples were filtered through
a 0.45 μm PET filter. The major cations (Ca, Mg and K) were measured through ICP-OES (iCAP 6300 duo, Thermo
Scientific). Before analysis, ICP samples were conserved using 1.5 mL (HNO3 69%) per 30 mL sample. TA was
determined using a SAN++ continuous flow analyzer (Skalar - NLD). The pH was measured using a HI3220 pH/ORP
meter. Dissolved organic carbon (DOC) and DIC were measured using a FormacsHT with LAS sampler (Skalar -
NLD). DIC and DOC were measured on eight and 12 occasions in leachates and pore water respectively.
**2.3 Soil collection and pretreatment**
Top soil pH was measured on five dates. To determine top soil pH, 4 g of air dried topsoil sample was dissolved in
10 mL DI water and shaken before pH measurement using a HI3220 pH/ORP meter (Hanna Instruments, Temse,
Belgium). After harvesting, soils were sampled using cylindrical soil cores (100 cm³, 5 cm length x 20 cm²). Samples
were taken across the depth of the mesocosm and three sampling depths were considered (0-20cm, 20-30cm, 30-
50cm). The cores were dried at 70°C for 48 hours to determine water content (gH$_2$O/g soil) and bulk density. An
additional soil sample was taken at each depth, dried at 70 °C for 48 hours and used for chemical analyses.





### 2.4 Sequential base cation extractions

As conceptualized by Tessier et al. (1979), base cations can reside in four different soil pools: the exchangeable

pool (where O-atoms on hydroxyl or carboxyl groups of clays or SOM associate with cations), the carbonate pool

(cations bound in pedogenic carbonates), the reducible pool (cations bound to Al/Mn/Fe hydr(oxide)) and the

oxidizable pool (cations bound to SOM). Organic matter bound to cations in these pools typically differs in stability;

SOM bound to cations in the exchangeable pool is expected to be more prone to microbial decomposition than

SOM bound in the oxidizable pool.

We adapted the original Tessier scheme by replacing 1M $MgCl_2$ with 1M NH4-acetate for extraction of the

exchangeable pool, in order to be able to measure all base cations in the exchangeable pool. Likewise, Na-acetate

was replaced with a mixture of acetic acid and water to be able to measure Na in the carbonate pool. We quantified

SIC changes from the base cations in these acetic acid extracts (as in (Larkin et al., 2022)) (see also Equation S4).

Additionally, three other SIC measurement techniques were explored to compare and the sensitivity of detecting

SIC changes after amending with a range of basalt (see section S3.7).

**Table 3:** overview of sequential extraction method
(extraction time, temperature, conditions, volume of extractants and chemical composition of extractants).

| Extraction scheme | Extraction scheme Adapted Tessier et al. (1979)* |
|---|---|
| Pool 1: Exchangeable pool | 10 mL 1M $NH_4(CH_3COO)$ 1h, 20°C, shaker → centrifuge → sample |
| Pool 2: Carbonate pool | 5 mL 1M acetic acid (2h, 20°C, shaker) + 4 mL $H_2O$ + 1 mL 3M NH4Acetate → sample |
| Pool 3: Reducible pool | 20mL 0.04M $NH_2OH.HCl$ in 25% (v/v) acetic acid (pH 2) 6h, 96°C, heat bath |
| Pool 4: Oxidizable pool | 3mL 0.02M $HNO_3$ +5mL 30%$H_2O_2$ (to pH 2 with $HNO_3$): 2h, 85°C, heat bath +3mL 30%$H_2O_2$ (to pH 2 with $HNO_3$) 3h, 85°C, heat bath +5mL 3.2M $NH_4(CH_3COO)$ (in 20 vol%HNO3) +4 mL $H_2O$ 0.5h, 20°C, shaker |

Prior to extractions, approximately 1g of soil was air dried. We also conducted the extractions for the pure basalt

that was initially added to the mesocosms to be able to correct for the cations that were initially already present as

exchangeable, carbonate, reducible and oxidizable pool cations. After each extraction, samples were centrifuged

for 2 minutes at 2000 rpm, supernatants were collected for analysis. The remaining soil pellet after centrifugation

was washed with 10 mL of demineralized water before the following step. Relevant elements (K, Na, Mg, Ca, Al,

Fe and Si) were measured in each pool using ICP-OES (iCAP 6300 duo, Thermo Scientific) for each pool. Si was

only assessed in the reducible pool and in the oxidizable pool to investigate whether Si forms amorphous oxides or





allophane-like compounds or binds with organic matter. Al carbonates were not quantified here as naturally these
carbonates are not commonly formed (Takaya et al., 2019).

**2.5 Plant responses**

On 26/8/2021(101 days after basalt amendment in soils), the aboveground biomass was harvested and dried  for
48h at 70 ºC to determine dry weight. Plant material was ground with an ultra-centrifugal mill (Model ZM 200, Retsch
GmbH, Haan, Germany). Base cations (Ca, Mg and K) were measured through ICP-OES (iCAP 6300 duo, Thermo
Scientific) in aboveground biomass to calculate plant base cation stocks. Base cations were measured separately
in all aboveground biomass parts: stems, leaves, flowers and corn ears.

**2.6 Calculation of Wr and CDR potential**

The Wr corresponds to the rate of rock dissolution. The Wr can be expressed per element or as moles of alkalinity
equivalents (i.e. the sum of base cation charges (**Equation 1**) per amount of rock surface area per unit of time (in
mol $m^{-2}$rock $s^{-1}$). We can use the TA increase in the system to calculate a 'CDR potential' (previously named
inorganic CDR equivalents by Vienne et al. (2023)). We define the CDR potential as the maximum inorganic CDR
that would be achieved if all base cations released during weathering that are currently retained in plants and soils
are exported to the ocean as alkalinity (**Equation 10).**
To calculate the Wr (from all base cation increases relative to controls in plants, extracted soil fractions and soil
water leachates), we sum changes in TA in the following pools: exported soil water (leachates), plants and soil
pools.We can express changes in the cation pool of each reservoir as the equivalent Wr required to supply the
cations ($Wr_{leachate}$, $Wr_{plant}$ and $Wr_{soil}$) (**Equation 2**). Conventionally Wrs are expressed using a logarithmic scale as
absolute values can vary strongly. We use a delta (Δ) to refer to differences relative to unamended control soil
throughout this work.

$$Log\ Wr\ \left[\frac{\Delta mol\ TA}{m^2 rock.s}\right] = Log\ (\frac{\Delta mol\ TA_{soil}\ +\ \Delta mol\ TA_{plant}\ +\ \Delta mol\ TA_{leachate}}{m^2 rock.s})\tag{2}$$

We used a gradient of rock applications, where we calculated the slope of the molar change in base cation charges
(expressed as an equivalent "alkalinity" change if these base cations were dissolved in water) with higher rock
amendment (TA slope) (**Figure 3**). We compared logarithmic and linear regression and selected the linear
regression approach, as both approaches had comparable R² and AIC values and linear slopes ease further data
processing (**Fig. S23 and Table S5**). We opted for the linear regression approach to simplify subsequent
calculations. To make our gradient approach more robust, we also calculated the log Wr for individual application
rates in **Fig. S13.**



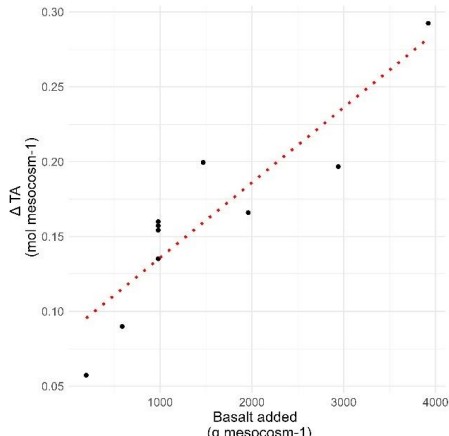


**Figure 3:** Illustration of the calculation of TA slope: The alkalinity scavenging by a given pool was plotted in function
of the applied basalt, after which the slope was used to quantify a Wr. This figure is an example regression with
data for the top soil exchangeable pool. All regressions can be found in **Fig. S23**.

Then we converted units of the alkalinity slope for increasing rock application (TA slope) per unit of rock mass to

moles of alkalinity per rock surface area and per time (**Equation 3**). Equation 3 was used to calculate $\Delta$mol TA m$^{-2}$

rock s$^{-1}$ of base cations in leachates, plants and of every measured soil pool at every soil depth.

$$\frac{\Delta mol\ TA}{m^2\ rock.s} = \frac{\textbf{Scavenged alkalinity (= TA slope)} \left[\frac{\Delta mol\ TA}{g\ rock}\right]}{\textbf{Experimental duration } [s] * SSA_{silicate} \left[\frac{m^2 surface\ area}{g\ rock}\right]} \tag{3}$$

$\Delta$mol TA m$^{-2}$ rock s$^{-1}$ was thus quantified per pool, based on the change in base cations in the basalt treatment

compared to the control treatment. For plants, we calculate TA slope through regression of harvested base cations

with basalt application. Harvested base cations were calculated as the product of harvested aboveground biomass

and their base cation content. Charge contributions of Na were not included; Na was not quantified at the time of

plant biomass elemental analysis, which may lead to an underestimation of the alkalinity equivalent increase in the

plant pool. However, given that base cation charges in the plant pool were about two orders of magnitude smaller

than in the soil pool, we expect the effect of this omission to be limited. In addition, maize plants aim to actively

increase their K/Na ratio which avoids salt stress, the K content of maize shoots is typically about 2 orders of

magnitude larger than Na (Gao et al., 2016; Suarez & Grieve, 1988). For leachates, TA slope was calculated as

the product of mean cumulative leachate volume and mean leachate TA concentration for each application rate and

regressing them with the applied basalt as dependent variable.

Finally, the Wr attributable to the change in cation content of the soil pools (Wr$_{soil.}$) was calculated by summing the

Wr$_{soil\_layer\_k\_pool\_j}$ for each pool and depth (**Equation 4**). Here, we sum changes in all pools of the topsoil (0-20 cm)

and lower depths (20-30 cm and 30-50 cm) to obtain an aggregate value for Wr$_{soil.}$. With



$Wr_{soil, layer\,k,\,fraction\,j}$ calculated as in **Equation 4** (with k = the number of depths and j = the number of considered
soil pools). TA slope at every depth and soil pool was calculated as in **Equation 5.**

$$Wr_{soil} = \sum_{k=1}^{3} \sum_{j=1}^{4} Wr_{soil, layer\,k,\,pool\,j} \qquad (4)$$

$$scavenged\ TA\ \text{(TA slope)} \left[\frac{\Delta mol}{g\ rock}\right] =$$

$$\frac{\frac{\mu mol\ TA}{g\ dry\ Soil}\ (Amended\ Soil) - \frac{\mu mol\ TA}{g\ dry\ Soil}\ (control\ Soil)}{Application\ rate\ (Amended\ soil)\ [g\ rock\ m^{-2}\ ground\ area] * 1000} * Bulk\ Density \left[\frac{kg\ dry\ Soil}{m^3 soil}\right] * thickness\ soil\ layer\ [m] \qquad (5)$$

TA per gram of dry soil mixture can be calculated for each mesocosm by summing the charges from each base
cation (**Equation 6**).

$$\frac{\mu mol\ TA}{g\ dry\ Soil} = \sum_{j=1}^{4} \left( \frac{\frac{\mu g\ Ca_{pool_j}}{g\ dry\ soil}}{40.078\frac{gCa}{molCa}} + \frac{\frac{\mu g\ Mg_{pool_j}}{g\ dry\ soil}}{24.305\frac{gMg}{molMg}} \right) * \frac{2mol\ TA}{mol\ cat^{++}} + \left( \frac{\frac{\mu g\ Na_{pool_j}}{g\ dry\ soil}}{22.990\frac{gNa}{molNa}} + \frac{\frac{\mu g\ K_{pool_j}}{g\ dry\ soil}}{39.098\frac{gK}{molK}} \right) * \frac{1 mol\,TA}{mol\ cat^+} \qquad (6)$$

These individual base cations (e.g. Ca in pool j) are calculated from the difference of cations weathered during the
weathering operation minus the cations that were already weathered initially in the feedstock rock (**Equation 7**).
For example, some cations can already exchange on the surface or edges of the applied minerals, so that these
cannot be counted as weathered, they will however contribute to CDR once leached.
To calculate in-situ Wr, it is thus necessary to correct for the cations that had already been weathered from primary
minerals at the time of silicate amendment. This correction is currently not being done in EW literature yet. As basalt
is only added to the top soil and not deeper, this correction is only done for the 0-20 cm soil layer here.

$$\frac{\mu g\ element_{i_{pool\,j}}}{g\ dry\ soil} = \left( \frac{\mu g\ element_{i_{pool\,j}}}{g\ dry\ soil} \right)_{Post\ weathering,\ soil\ mixture} - \left( \frac{\mu g\ element_{i_{pool\,j}}}{g\ dry\ soil} \right)_{added\ with\ feedstock\ initially} \qquad (7)$$

The mass of a specific element (i) in each of the four (j) soil pools (in μg element/g soil) is calculated using **Equation**
**8.**

$$\left( \frac{\mu g\ element_{i_{pool\,j}}}{g\ dry\ soil} \right)_{Post\ weathering,\ soil\ mixture} = \frac{concentration\ element_i\ in\ pool_j\ \left[\frac{mg}{L}\right] * Volume\ extract_j[mL]}{mass\ of\ solid\ extracted\ [g]} \qquad (8)$$

The initial addition of element i to pool j is calculated as in **Equation 9**.

$$\left( \frac{\mu g\ element_{i_{pool\,j}}}{g\ dry\ soil} \right)_{added\ with\ feedstock\ initially} = \frac{\mu g\ element_i\ pool_j}{g\ silicate} * \frac{Application\ rate\ \left[\frac{g\ silicate}{m^2}\right]}{Bulk\ density\ \left[g\ dry\frac{soil}{m^3}\right] * depth\ of\ soil\ amendment\ [m]} \qquad (9)$$

According to the charge balance (**Reaction 1-3**) during mineral dissolution, 1 mol $HCO_3^-$ $mol^{-1}$ TA is generated (and
thus 1 mol of $CO_2$ is sequestered). We define a factor η, that is equal to the ratio of $HCO_3^-$ per mol of generated
TA. According to charge balance, η=1. A more conservative approach is to assume that all this generated alkalinity
will be exported to the ocean, after which chemical equilibrium degasses a portion of the alkalinity (η = 0.7 mol $CO_2$
$mol^{-1}$ TA, assumed for oceans) (Renforth et al., 2012; Renforth et al., 2019; Renforth & Henderson, 2017).



According to Renforth et al. (2019), the ocean alkalinization efficiency η ranged between 0.7 and 0.85. This η
parameter is relatively uncertain given that model studies indicate that η can range between 0.65 and 0.8 mol CO2
mol TA$^{-1}$ (see section S6 in the supplement of (Katarzyna et al., 2024)). Alternatively, we can assume that all base
cations will form solid carbonates in soils or rivers. In this case η=0.5 mol $CO_2$/mol TA (**Reaction 4**). In **Table 4**, we
calculated CDR potentials assuming conservative values of η=0.5 (carbonate precipitation scenario) and η=0.7
(lower boundary of ocean alkalinization efficiencies considered by Renforth et al. (2019)).
While cations added with the rock feedstock were subtracted to calculate Wr (**Equation 7**), they are not subtracted
to calculate the CDR potential as scavenged TA in rock feedstock can also leach to soil water whereby $HCO_3^-$ is
generated.  Last, base cation changes in the plant pool were excluded from the CDR potential pool here, as a
conservative approach we assume that base cations in plants will not reach the ocean. The latter assumption had
a negligible impact on the CDR potential estimate (**Table 4**).
$$CDR\ potential\left[\frac{kg\ CO2}{t\ rock}\right] = Scavenged\ TA\left[\frac{mol\ TA}{g\ rock}\right] * \frac{\eta\ molCO2}{mol\ TA} * \frac{44gCO2}{mol\ CO2} * 1000 \tag{10}$$
$$with\left(\frac{\mu g\ element_{i_{pool\ j}}}{g\ dry\ soil}\right)_{added\ with\ feedstock\ initially} = 0$$


**2.7 Calculation of the carbonate saturation indices (SIc) using Phreeqc**
To assess whether carbonate precipitation was theoretically possible during this experiment, we calculated SIc for
dolomite and calcite. For Mg and Ca the SIc as the logarithm of the ion activity product and the solubility product
constant if dolomite and calcite (SIc = log IAP/K). Minerals tend to precipitate when a log SIc >0 is reached. Likewise,
they are in perfect equilibrium at a log SIc =0 and tend to dissolve if log SIc <0. The R phreeqc package was used
and the phreeqc.dat database was used. As an input, the experimental pore water (10 cm depth) composition of
Mg and Ca was entered, as well as measured pH and TA. Daily SIc values were calculated by feeding unique
combinations of Mg, Ca, pH and into the PHREEQC solution function for each day.
**2.8 Data analysis**
For SIC and elemental stocks in plant biomass, soil pools and soil water export, a linear regression with basalt
application rate as a dependent variable was performed to test for a basalt effect. For measurements that were
repeated in time (pore water and leachate DIC and DOC compositions), a linear mixed model was used with basalt
and time as fixed factors and mesocosm as a random factor using the lme4 R package (version 1.1-33). For
measurements repeated in time, we assessed basalt x time interaction effects and discarded these if not
significant.All analyses were executed in R version 4.3.2. As an additional sensitivity analysis for the determination
of Wr using the slope of application rates approach described in the main text, we quantified Wr also for individual
application rates in Fig. S13.



## 3 Results

Basalt amendment significantly increased DIC and TA in the top soil water (**Figure 4**). TA in soil water correlated

positively with DIC (R² = 0.68, p<0.01, **Fig. S10**). TA was thus generated in the basalt amended soil layer, yet we

did not observe DIC or TA increases with higher basalt application rates in water exported from the soil column

(**Figure 4**). Temporal dynamics show that DIC in top soil pore water gradually increased in time with higher basalt

amendment, while DOC decreased in time with more basalt (**Fig. S7 and Table S4)**.

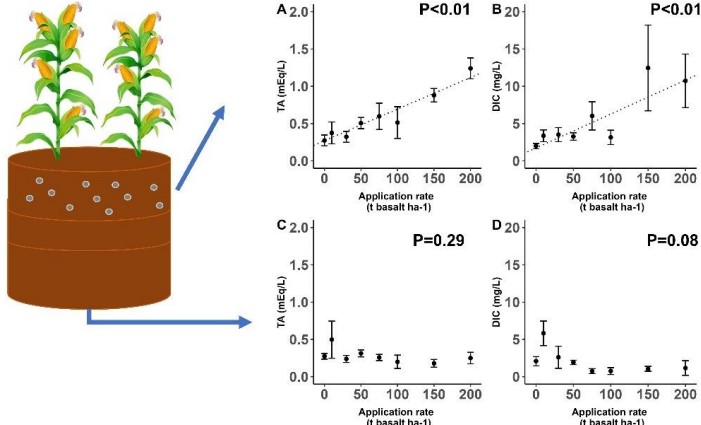

**Figure 4:** Top soil (0-20 cm) pore water (A) TA and (B) DIC. Export water (50 cm depth) (C) TA and (D) DIC. Values
represent average concentrations +- SE across all sampling occasions over the 100 day experiment. Significant
trends are indicated with a dotted regression line.

Overall, base cations were mostly retained in the top soil, where Ca significantly increased in the exchangeable,

reducible and oxidizable pools with higher basalt addition. Only in the carbonate soil pool, Ca (and also Mg)

significantly decreased with more basalt (**Figure 5 and 6**). With higher rock amendment, Mg accumulated in the

top soil exchangeable pool (p<0.01) and gave an even larger, yet borderline significant (p=0.07) signal in the

reducible pool. Changes in Na followed similar patterns as Mg, as also significantly more Na exchanged in top soil

(p=0.02) and a larger signal of reducible Na was found (p<0.01). In contrast with divalent cations, monovalent

cations increased in the carbonate fraction if basalt increased (**Figure 5 and 6**). With more basalt, Al formed both

reducible and oxidizable compounds in top soils, while Si increased only significantly in the oxidizable pool (p=0.04)

(**Figure 5 and Fig. S15**). Increases in oxidizable Si, Ca, Al with higher basalt addition suggest the formation of

mineral-associated organic matter.

 In the soil layer just below the soil-basalt mixture (20-30 cm), the cations did not increase significantly in any of the

measured soil pools and oxidizable Na, Fe and Mg even decreased significantly (**Fig. S11, Figure 6**). We did not

observe significant changes in any element with higher basalt amendment in the 30-50 cm soil layer (**Fig. S12).**




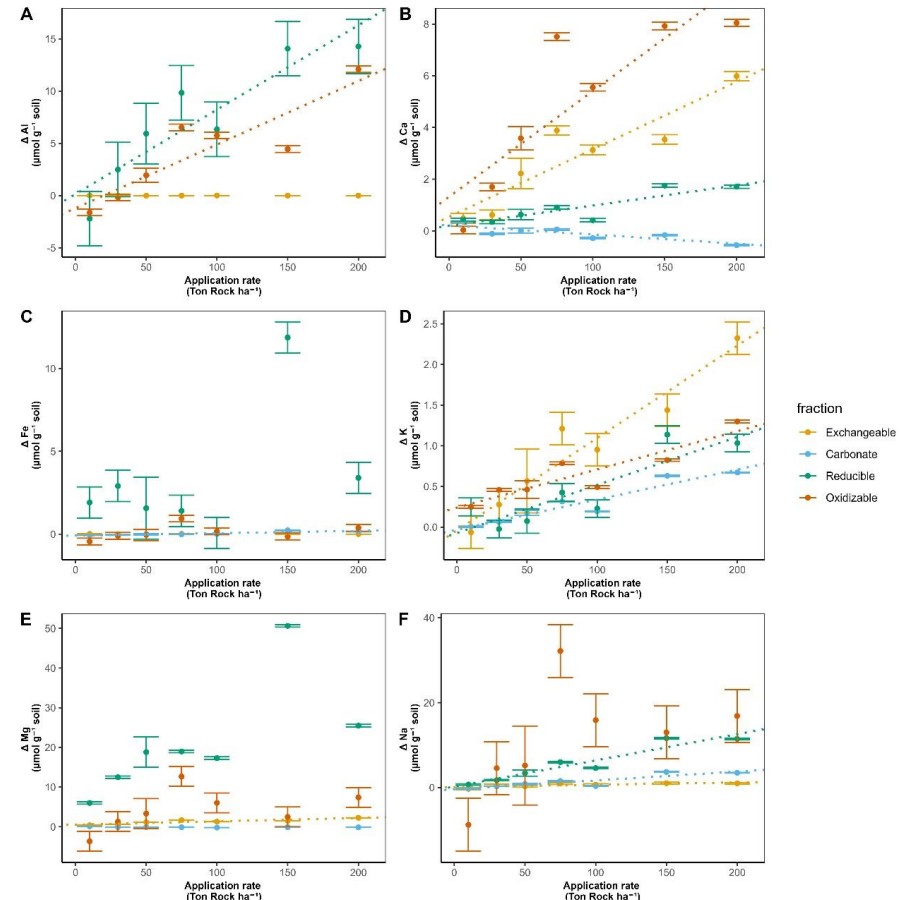

**Figure 5:** Change in top soil (0-20 cm) elements relative to the control soil (corrected as in **Equation 7**), 101 days
after basalt amendment, as a function of basalt application rate for (A) Al (B) Ca (C) Fe (D) K (E) Mg and (F) Na for
four different soil pools. Dots and error bars represent averages and SEs. Significant effects (p<0.05) of basalt
application rate on cation concentrations are indicated with dotted linear regression lines. Note that y-axes absolute
values differ for subplots to visualize smaller changes for certain elements. This data was normalized for control
soil concentrations, raw data that is not normalized for control soil contribution can be found in **Fig. S19-21.**


From all significant element changes in soil pools, we calculate that 8.4%, 52.1% and 9.4% Of basalt Na, K and Ca
were weathered while we do not observe an increase in Mg if we only consider significant (p<0.05) slopes. If we
consider all (also p>0.05) regression slopes, the estimates become  48.0%, 10.6%, 9.35% for K,  Ca and Mg, while
Na did not increase in this approach (mass balance per element, see **Fig. S24).**

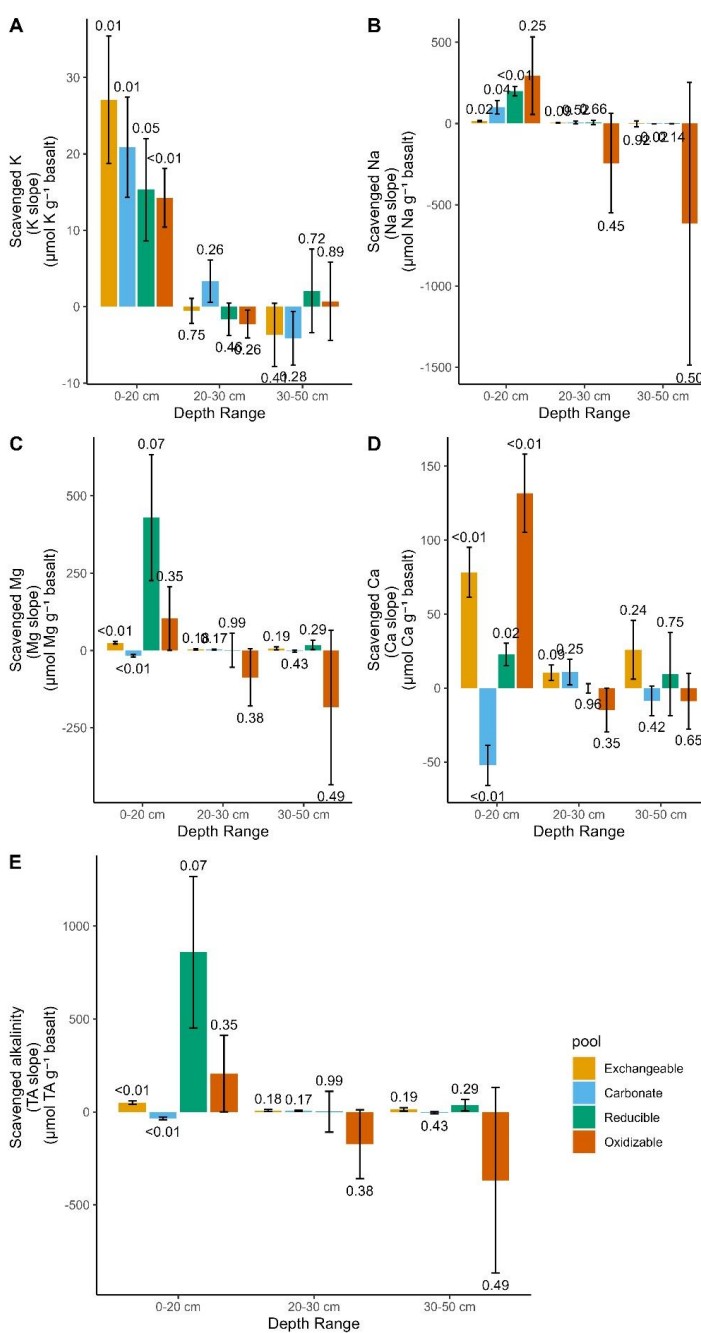

**Figure 6:** Equivalent alkalinity uptake 101 days after basalt amendment in different soil pools and depths. P-values of linear regressions are shown above and below bar plots of positive and negative changes respectively. Error bars represent the SE of the mean.



Base cations were not only scavenged by soils, but also by plants. Although two orders of magnitude smaller than
in soil pools, TA scavenging by plants was higher than soil water exported TA and increased significantly with larger
basalt amendment (p<0.01) (**Figure 7**). The increase in base cation charges in plants was attributed to K (81%),
Ca (11%) and Mg (8%).

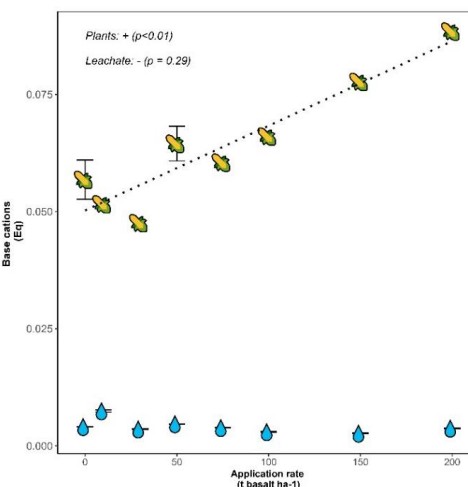

**Figure 7:** Moles of base cation charge equivalent (Eq) per mesocosm after 101 days retained in maize plants (stems, leaves and corn ears) (indicated with maize fruit symbols) and) flushed with leaching water (indicated with droplets). Error bars represent averages and SEs. Note that Na was not analyzed in plants and is thus not included in the harvested base cations. Errors on leachate TA were small and appear as horizontal lines within the droplets.

Converting the base cations to moles of equivalent TA and considering only the exchangeable pool as only soil
cation reservoir we derive a log weathering rate of $-12.13\pm 0.34$ mol TA m$^{-2}$ rock s$^{-1}$ (**Table 4**). When we consider
also the decrease in base cation equivalents in the carbonate pool, the mean estimate decreases to $-12.23$ mol TA
m$^{-2}$ rock s$^{-1}$ , translating into mean estimated CDR potentials of 0.36-0.5 kg $CO_2$ t$^{-1}$ basalt (assuming $\eta$=0.5-0.7). If
we include all soil pools and non significant regressions the estimates becomes one order of magnitude higher, yet
with substantial uncertainty.
**Table 4:** Overview of the Wr and CDR potential that can be quantified from changes in base cations in specific soil pools.
Rows where (scavenged) TA increased significantly with increasing basalt amendment are indicated in bold.

| Soil Pool | Depth | Reservoir | Log Wr (Log mol TA/m² basalt /s) | CDR Potential* (kg CO2/ton basalt) ($\eta$ = 0.5) | CDR Potential* (kg CO2/ton basalt) ($\eta$ = 0.7) |
|---|---|---|---|---|---|
| / | / | Plant* | **-12.93±0.07** | / | / |
| / | / | Leachate* | **Wr<0** | / | / |
| Exchangeable | 0-20cm | soil | **-12.20±0.41** | **1.10±0.04** | **1.55±0.05** |
| Carbonate | 0-20cm | soil | **Wr<0** | **-0.75±0.03** | **-1.05±0.04** |
| Reducible | 0-20cm | soil | -10.96±0.21 | 18.91±1.76 | 26.47±2.46 |
| Oxidizable | 0-20cm | soil | -11.58±0.43 | 4.53±0.89 | 6.34±1.24 |
| Exchangeable | 20-30cm | soil | -13.02±0.29 | 0.17±0.02 | 0.23±0.03 |
| Carbonate | 20-30cm | soil | -13.17±0.29 | 0.12±0.02 | 0.16±0.02 |
| Reducible | 20-30cm | soil | -13.17±50.43 | 0.02±0.47 | 0.03±0.66 |
| Oxidizable | 20-30cm | soil | Wr<0 | -3.82±0.80 | -5.35±1.12 |
| Exchangeable | 30-50cm | soil | -12.76±0.30 | 0.30±0.04 | 0.42±0.06 |
| Carbonate | 30-50cm | soil | Wr<0 | -0.10±0.02 | -0.15±0.03 |
| Reducible | 30-50cm | soil | -12.34±1.15 | 0.79±0.13 | 1.11±0.19 |
| Oxidizable | 30-50cm | soil | Wr<0 | -8.10±2.15 | -11.34±3.01 |
| Exchangeable | 0-20 | Exchangeable+plant+leachate | **-12.13±0.34** | **1.30±0.04** | **1.82±0.05** |



| Exchangeable + Carbonate | 0-20 | Exchangeable+plant+leachate+ carbonate | **-12.23±1.05** | **0.36±0.05** | **0.50±0.07** |
|---|---|---|---|---|---|
| All soil pools | All | All soil pools + plant + leachate | -11.11±2.70** | 13.17±3.07 | 18.43±4.29 |

*For leachates (which represents realized CDR) and also for plants there is no CDR potential in this approach.
** Abs (Wr/standard error (Wr)*LN(10)) was used to propagate the error using the log10 transformation, resulting
in substantial uncertainty for the Wr estimate of all pools.

## 4. Discussion

### 4.1 Weathering rates and CO$_2$ removal

EW is typically considered as a durable CDR pathway that removes CO$_2$ from the atmosphere by producing DIC that is either transported to the ocean (Strefler et al., 2018) or precipitates as carbonates in the soil (Manning et al., 2013). Here, we observe a clear weathering signal (a TA and DIC increase) in top soil pore water (**Figure 4**). These TA and DIC increases in the pore water of amended top soil are consistent with recent findings (Holzer et al., 2023; McDermott et al., 2024; Vienne et al., 2024). DIC did however not leach from our soil columns within this experimental timeframe of 101 days.

Absence of substantial DIC leaching is in line with other short-term recent studies (Amann et al., 2020; Larkin et al., 2022; Niron et al., 2024; Vienne et al., 2024). For example, DIC export after 1 year in a mesocosm trial with 220 ton ha$^{-1}$ olivine-rich rock was about three orders of magnitude lower than what would be expected from lab-scale weathering shake flask studies (Amann et al., 2020). Vienne et al. (2024), amended soils with 100 ton basalt ha-1 and quantified that CDR from exported TA that was in the same order of magnitude as in the work of Amann et al. (2020). Although the studies of Amann et al. (2020), Vienne et al. (2024) were relatively short (<= 1 year) and used a relatively low water infiltration flux, also a longer (3 year duration) catchment-scale study in Malaysian oil palm plantations with high annual rainfall (>2000 mm year$^{-1}$) detected no significant increase in TA leaching in the catchments (Larkin et al., 2022).

This DIC leaching delay can have multiple causes (**Figure 1**); A first possibility is pedogenic carbonate formation. We observe that solid carbonates did not increase in our experiment, SIC even decreased in time. PHREEQC calculations for our experiment suggest that dolomite and calcite were undersaturated, so that carbonate dissolution was possible (**Fig. S17**). Saturation states are expected to be low in our experiment because control soil was undersaturated and dissolved base cations were scavenged by other soil pools (**Figure 5 and 6**). A decrease in SIC is in contrast with substantial SIC increases found after wollastonite rock amendment (Haque et al., 2019, 2020). For short-term basalt studies, using elemental C analysis, also no significant changes in SIC could be detected previously (Kelland et al., 2020; Vienne et al., 2022, 2024). In contrast, in the study of Larkin et al. (2022), a relatively small SIC increase was detected in amended soils, using carbonate pool extractions.





While TA was not exported or taken up by soil carbonates here and plant base cation losses were minor (**Table 4**)
it was retained in top soil where the exchangeable and pools reduced solute TA. Our log $W_r$ estimate quantified
from significant changes in TA uptake with higher basalt amendment only was approximately -12 mol TA m$^{-2}$ s$^{-1}$.
with basalt in soils measuring base cation scavenging only in the exchangeable pool (Kelland et al., 2020;
Reershemius et al., 2023; Reynaert et al., 2023; te Pas et al., 2023), where log $W_r$ was between -12 and -11 (see
Table summarized by Vienne et al., (2024)). Estimates from Buckingham et al. (2022), based only on leachates,
gave a much lower log $W_r$ of -15, partly due to low water infiltration rates. Even with a high infiltration flux (8000
mm/year), Amann et al. (2022) estimated log $W_r$ between -12.5 and -13.5 from basalt leachates. This highlights the
importance of including scavenged alkalinity to determine $W_r$ in soils. When we also include non-significant
regression slopes we derive a mean log $W_r$ estimate with substantial uncertainty (-11.11±2.70) mol TA m$^{-2}$ s$^{-1}$. From
individual application rates, we even quantify log $W_r$ ranging between -11 and -10 (**Fig. S13**); these values are
comparable to basalt in soil-free shake flask experiments at circumneutral pH (Brantley et al., 2008).

Although this and other experiments quantify a relative consistent weathering rate from exchangeable bases and
derived rates are comparable to shake flask experiments, we emphasize that unlike in shake flask experiments
where base cations remain irreversibly dissolved, in soils, solid-phase base cation scavenging causes DIC
degassing **(Figure 1)**. From the sum of significant TA slopes we calculate a relatively low CDR potential, equalling
to only approximately 0.4-0.5 kg $CO_2$ ton$^{-1}$ basalt or 0.020-0.025 t$CO_2$ ha$^{-1}$ for a basalt application rate of 50 t ha$^{-1}$
(**Table 4**). We emphasize that a CDR potential is a maximum inorganic CDR that can be realized with the delivered
amount of base cation weathering as strong acids associated with fertilizers (such as nitric acid and sulphuric acid),
or organic acids and not carbonic acid may have initially weathered silicate rock which does not lead to a CDR
(McDermott et al., 2024; Taylor et al., 2020).

For climate change mitigation, not only the amount of CDR potential is important, but also the timescale at which
this CDR is realized (Kanzaki et al., 2024). A mass balance of TA shows that exported TA was negligible compared
to scavenged TA that was retained in the soil (**Table 4**). As long as TA is retained in soil pools, inorganic CDR
through DIC export is delayed as equivalent amounts of protons have then been released into the soil water to
maintain charge balance (**Figure 1**). Realization of this delayed inorganic CDR depends on liberation of base
cations from these soil pools and their transport out of the soil, charge-balanced by $HCO_3^-$. This export may take
decades or longer, depending on the circumstances (Kanzaki et al., 2024).

The realization of CDR may be even further delayed through the formation of base cation bearing clay minerals.
Clay formation has previously been suggested for EW application based on changes in soil water Ge/Si ratios and
Si isotopes (Vienne et al., 2024). These measurements indicated basalt induced clay formation, but it remains





unclear what type of clays were formed and hence what the effect on inorganic CDR may be. In the best case for
the inorganic CDR lag, the formed clays are 1:1 phyllosilicates such as kaolinite and do not have base cations. In
this case, DIC leaching is only retarded by base cation exchange. Worst case for the inorganic CDR time lag, the
formed secondary minerals bear substantial amounts of base cations such as chlorite or chrysotile. These clays
exhibit a log Wr between -12 and -12.5 at neutral pH (Palandri & Kharaka, 2004), so that dissolution within decadal
timescales is unlikely (Bullock et al., 2022).

To investigate whether base cation bearing clays could be forming in the top soil reducible pool in this experiment,
we compared Mg/Si and Al/Si ratios with common clays (**Fig. S22**). We could not find a good stoichiometric match
between reducible pool and known crystalline clay phases. Still, amorphous clay precursors with deviating
stoichiometry could be present in the reducible pool and crystalline clays could also be hiding in the unassessed
residual soil pool that remains after the sequential extraction procedure (Niron et al., 2024). Ryan et al., (2008)
showed that <20% of crystalline clay minerals can be extracted with the similar BCR extraction scheme and an
additional aqua regia digest is required to measure clays. We therefore suggest for future research to add an
additional clay targeting leach.

Although unfavourable for inorganic CDR, if base cation bearing secondary clay minerals would form, they can
increase SOC (Georgiou et al., 2022; Heckman et al., 2022). Georgiou et al. (2022) refers to the latter clays as
'high-activity minerals' due to their higher SOM stabilization capacity compared to secondary minerals that do not
contain base cations (i.e., 'low-activity minerals', with a lower CEC such as kaolinite).  Both high- and low-activity
minerals can adsorb DOC and form mineral-associated organic matter-C (MAOM-C), which is believed to have a
relatively high permanence (decades-centuries) in soils (Lavallee et al., 2020). Besides mineral surface however,
plant inputs can also limit SOC accrual. In the latter case, SOC stocks can only increase if belowground plant C
inputs increase, which could follow from increases in exchangeable bases or pH (Haque et al., 2019; Shamshuddin
et al., 2011). Nonetheless, increases in decomposition can also stimulate SOC losses if rock dust increases soil pH
(Klemme et al., 2022). The response of SOC to EW is thus prone to several contrasting mechanisms and requires
further investigation.



**4.2  Implications for monitoring inorganic CDR**





Different base cation monitoring strategies are possible. A first option is  to quantify TA in soil water (Isometric,
2024). A disadvantage is however that soil water samples have to be sampled across the soil depth. Alternatively,
TA could be only monitored in top soil, yet then uncertain TA leaching models must be used (Kanzaki et al., 2024).
To decrease the uncertainty of TA leaching models, soil measurements in depth profiles could be used to calibrate
these models.

A first soil measurement approach is a mobile/immobile element approach, which tracks cation losses from
amended top soils (Reershemius et al., 2023). However, this approach only focuses on the top soil and fails to
account for cation loss from top soils due to erosion or vertical feedstock transport via infiltration or bioturbation
(Reershemius et al., 2023). In addition this approach does not track potential TA scavenging by organic matter or
clay formation at larger depth.

Alternatively, entire depth profiles could be analyzed to spatially calibrate TA leaching models. The Isometric
protocol already includes the analysis of the exchangeable soil pool as a requirement. Adding also the carbonate,
reducible and oxidizable soil pools to the analysis could make base cation mass balancing more complete. These
protocols could calibrate predictive TA leaching models spatially. In addition there is an opportunity to quantify soil
organic carbon (SOC) and MAOM-C changes in the same samples, which have recently gained traction in EW
research due to their role in stabilizing SOM (Buss et al., 2024; Sokol et al., 2024; Xu et al., 2024). Integration of
these measurements can provide more accurate estimates of the climate impact of EW, but should take into account
the difference in permanence of inorganic and organic carbon stocks.
However, this MRV approach involves complexities such as feedstock correction, leaching solution strength and
soil heterogeneity. Although correcting for pre-weathered elements was crucial in this study, it assumes perfect
mixing based on a silicate-to-soil ratio. This correction was particularly significant for carbonate and reducible soil
pools, where for some base cations, over half of the cation increase with basalt amendment originated from
feedstock addition and not from weathering (**Fig. S18**). An alternative approach could involve creating time series
from sequential extraction data and quantifying base cation changes based on the slope between two
measurements taken after rock amendment.
As discussed in the previous section, another key challenge is that the fate of base cations may remain uncertain
if strongly bound crystalline secondary minerals form that are unextractable by the Tessier scheme. Pogge von
Strandmann et al. (2022) proposed substituting the $H_2O_2$ leaching step of the Tessier scheme with a dilute HCl
leach, which is thought to extract clays as well. Alternatively, post-extraction analysis of residual solids using
techniques such as XRD or QEMSCAN may be necessary to rigorously assess changes in rock mineralogy (Mason
et al., 2022). Although deep soil core sampling and extensive mineralogical analysis are resource-intensive and not



feasible for large-scale application, this monitoring strategy could be valuable during the initial adoption of EW in
targeted 'measure-all' experiments, as reliable TA leaching models require extensive calibration.

## 5. Conclusions

This study presents a detailed examination of EW and its effectiveness as a climate mitigation technique, revealing
both its potentials and limitations. A novel aspect of this work is the in-depth investigation of entire soil profiles for
base cations in different soil fractions, paired with soil water TA monitoring. We highlight the value of sequential
extractions as a method for monitoring base cations throughout soil profiles for calibrating TA leaching models.
Our results suggest that EW using basalt amendments may not yield the immediate inorganic carbon dioxide
removal (CDR) benefits previously anticipated. We observed rock weathering without inorganic CDR; despite the
absence of DIC leaching or carbonate precipitation, exchangeable bases increased with higher basalt amendments,
proving that rock weathering occurred. Additionally, we observed a borderline significant yet substantial increase in
reducible bases in top soils with more basalt, which may further retard TA leaching.
As base cation exchange increased with higher basalt amendments, we infer that the eventual release of DIC from
soil minerals into surface waters will be further delayed with higher rock applications. This renders high basalt
application rates less effective as a strategy for achieving rapid inorganic CDR. It remains unclear if clays were
formed here and whether EW can deliver CDR within the urgent decadal timeframe needed to mitigate climate
change. Despite its limitations for short-term inorganic CDR, the generated secondary minerals and increased
cation exchange capacity (CEC) could enhance plant productivity and soil organic carbon (SOC) retention in soils,
contributing to long-term soil health, fertility, and potentially carbon sequestration beyond inorganic pathways.

## Acknowledgements

We thank Anne Cools, Steven Joosen and Anke De Boeck for their assistance with ICP-OES for sequential
extraction samples and Anthony De Schutter to characterize basalt using XRD. We thank DURUBAS to provide
basalt and provide the XRF material data sheet. We thank Tom Cox for fruitful discussions. We acknowledge the
use of Microsoft Copilot to improve the English of this manuscript. This research was supported by the Research
Foundation— Flanders (FWO) [1S06325N], 1174925N] and [G000821N] (Biotic controls of the potential of
enhanced silicate weathering for land-based climate change mitigation). We also acknowledge support of the
UPSURGE project, which has received funding from the European Union's Horizon 2020 research and innovation
program under grant agreement No 101003818.

## Author contribution

AV: research conceptualization, data gathering, development methodology, data analysis and writing. PF: conceptualized
sequential extraction methodology, writing and discussion. JR: research conceptualization, data gathering. TJS: writing and
discussion. TR: writing and discussion. RP: data gathering, rock characterization and writing. JH: writing and discussion. HN:
development extraction methodology, writing and discussion. MPE: elemental C measurements and proofreading. LS: writing,
development methodology and discussion. LB: writing and discussion. SV: supervising research, conceptualization, writing and
discussion.




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
