# Peer review of "Weathering without inorganic CDR revealed through cation tracing."

_EGUsphere, 2025_

## Author Comment (AC2)

**Reviewer1):**

Q0) Main points: "Solid soil pools". Analysis of different soils fractions to assess potential for uptake of weathered base cations is a key focus of the study. This needs better introducing, better assessment and better quantification of uncertainties.

We thank the reviewer for the detailed and constructive comments. We have improved the introduction of the different soil fractions (see Q1), discussed the composition of these soil pools (see Q3) and clarified the meaning of uncertainties in all figures (see Q2).

Q1) In the introduction, expand the paragraph starting on L78 to include a discussion of how the reducible and oxidisable fractions take up base cations (specifically Mg, Ca, K and Na) and provide key references from the literature that provide evidence for this (and presumably motivated the approach used in this study).

We agree with the reviewer's suggestion to elaborate the introduction with more insights on the reducible and oxidizable fractions. For the reducible pool, the best known mechanism is the adsorption of MgO complexes on hydrous ferric oxides. The adsorption of base cations by organic functional groups in the oxidizable pool is best described by Tipping & Hurley.

We expanded the introduction with this information.

Line 100: Thirdly, Tessier et al. (1979) operationally defined a reducible soil pool where base cations are associated with iron (Fe) and manganese (Mn) (hydr)oxides, and an oxidizable pool where cations are bound to SOM or sulfides. Dzombak and Morel (1990) modelled adsorption of Mg to hydrous ferric oxides (FeO(OH)), in which a surface hydroxyl group loses a proton and is replaced by a magnesium ion (FeOOH + Mg2+  $\Leftrightarrow$  FeOMg+ + H+) and thereby decreases solute TA. In the fourth considered soil pool, the oxidizable pool, organic functional groups such as carboxylic and phenolic groups can form strong bounds with cations after deprotonation (Kalinichev et al., 2011). Cations in the oxidizable pool are expected to chemically stabilize organic matter due to cation bridging and inhibition of decomposing enzymes (Rowley et al., 2018).

Q2) Multiple (n=??) separate analyses of the same soils and/or multiple experiments (at least for the control and 50 t applications)? The reproducibility of leaching needs to be reported- it is not clear how the error bars shown in for example Fig. 5 were generated. Was uncertainty associated with control soil and basalt feedstock propagated?

We apologize for the confusion regarding the replicates. Our experiment consisted of a treatment gradient with one replicate per treatment. To capture experimental variance, we included five replicates for two treatments: the 0 t/ha and 50 t/ha treatments. This and propagation of uncertainty was better clarified in the revised manuscript, including the figure legends.

Averages and standard errors for every replicated application rate (0 or 50 t/ha) were determined. The average from the 50 t ha-1 was subtracted with the average from the control soil and se =  $\sqrt{\text{(se\_control^2 + se\_basalt^2)}}$ . For non-replicated application rates (10,30, 75,100, 150 and 200 t/ha, se=0) the measurement was subtracted from the control soil average and errors were also propagated with se =  $\sqrt{\text{(se\_control^2 + se\_basalt^2)}}$ .

We added an explanation for the error propagation in figure 5 in the methods:

Line 357: To propagate uncertainty between basalt and controls in Figure 5, averages and standard errors for every replicated application rate (0 or 50 t/ha) were determined. The

average from the 50 t ha-1 was subtracted with the average from the control soil and se =  $\sqrt{\text{(se\_control}^2 + \text{se\_basalt}^2\text{)}}$ . For non-replicated application rates (10,30, 75,100, 150 and 200 t/ha, se=0) the measurement was subtracted from the control soil average and errors were also propagated with se =  $\sqrt{\text{(se\_control}^2 + \text{se\_basalt}^2\text{)}}$ .

For figure 5, for example, the legend will for example be revised as follows:

**Figure 5:** Change in top soil (0-20 cm) elements relative to the control soil (corrected as in **Equation 7**), 101 days after basalt amendment, as a function of basalt application rate for (A) Al (B) Ca (C) Fe (D) K (E) Mg and (F) Na for four different soil pools. Dots and error bars represent averages and standard errors. For basalt application rates other than 50 t ha-1, error bars correspond to those of the control soils, as these basalt treatments were not replicated and the data are shown as control-normalized results. Significant effects (p<0.05) of basalt application rate on cation concentrations are indicated by dotted linear regression lines. Measurements were repeated on at least four samples per fraction for the control soils (N ≥4 for each fraction) and N=4 for 50 t ha-1 treatment (fewer than 5 replicates were available due to technical issues). Note that y-axes scales differ among subplots to better visualize small changes for some elements. Unnormalized (raw) data are presented in **Fig. S19-21.**

We also clarified the figure labels of all other figures with error bars in the figure labels.

Q3) Chemical leaching techniques are notoriously unreproducible and may attack other phases in addition to the target phases. There also needs to be assurance as to what phases were released- how do the authors know that the target phases were indeed those released? The authors seem to doubt this themselves in L473-480. Basalt and soil will likely respond very differently to chemical leaching, yet it is assumed they respond the same. Related, the authors then assess whether the soil reducible pool corresponds to clays (L473)- is the reducible fraction is Fe- and Mn-(oxyhydr)oxides (as stated in L172, or not? The paragraph L473-480 undermines the rest of the paper- saying more research is needed is not adequate (it is something I expect to read in an undergraduate dissertation, not a manuscript submitted for peer-review publication).

We agree with the reviewer that these leaching techniques are not fully specific may attack non-target phases. Base cations in the reducible pool are indeed typically adsorbed to Fe- and Mn-(oxyhydr)oxides (e.g. MgO- adsorption on hydrous ferric oxides is well known, see e.g. Dzombak and Morel (1990)). It is also known that the chemical extraction agent used for the reducible pool (hydroxylamine) can attack crystalline clay phases such as nontronite (see Figure 3 from Ryan et al. 2008). Similar artefacts can occur with H2O2 (used for the oxidizable pool extractions) (also Figure 3, Ryan et al. 2008). Moreover, the elevated Si in the topsoil reducible and oxidizable pools (**Fig. S15**) also suggest that we are extracting more than Fe-/Al-hydroxide adsorbed base cations and that the extraction of clays was likely.

To address this, we revised the paragraph and strengthened the point of potential clay formation. In the revised text, we now clarify that we mainly expect cations associated with Fe- and Mn-(oxyhydr)oxides in the reducible pool and with organic matter in the oxidizable pool based on literature, but due to the poor specificity of the extraction agent also crystalline clay minerals may have been extracted (see Ryan et al., 2008)

Following the reviewer comment, we also deleted the comparison of extracted stoichiometry with clays ((and the associated figure), as this comparison may not be meaningful given that the extracted material likely represents a mixture of oxides and clays.

While we cannot unambiguously identify the specific secondary phases formed, our results clearly indicate that substantial base cation retention occurs in non-exchangeable soil fractions. This finding is robust and supports our main conclusion that base cation losses to soils extend beyond the exchangeable pool, with implications for inorganic carbon removal efficiency.

**Line 500:**

We expect cations to be primarily associated with Fe- and Mn-(oxyhydr)oxides in the reducible pool and with organic matter in the oxidizable pool, as supported by literature (Tessier et al., 1979); However, the extraction reagents of this sequential extraction scheme (hydroxylamine and  $H_2O_2$ ) are known to have a limited specificity and may have also partially targeted other mineral phases (such as clays)(Ryan et al., 2008), which could explain the elevated Si observed in the topsoil pools (**Fig. S15**). In addition, the observed increase of aluminum in association with the reducible soil fraction indicate the formation of secondary minerals. While we cannot pinpoint the exact Mg-phases formed in our soils, our results clearly indicate substantial base cation retention in the soil beyond the exchangeable pool.

We also refer to the answer on Q39 for discussion of this point.

Q4) Terminology. "Inorganic CDR" is used widely (even in the title!) but is not defined. Similarly, "CDR potential" should be explicitly defined and these terms should be compared with those in the literature (expanded under Introduction below). There is massive confusion around the definition of CDR and differences in quantification approaches used by the EW community, and this paper as it stands only adds to that. It's really important to sort this out, to avoid accusations of green-washing and worse. If the term "inorganic CDR" is to be used, it needs to be defined, have expertise in the field but I am not familiar with this terminology. See comments below on soil pools.

We thank the reviewer for this important remark regarding terminology and specifically the use of inorganic CDR and CDR potential. We have carefully revised the terminology in the manuscript in this regard, now following the terminology of Steinwidder et al (2025), a recently published study that used comparable methodology. Steinwidder et al (2025) distinguished realized and potential inorganic CO2 removal. In the revised manuscript, this implied the following changes:

First, we replaced inorganic CDR anymore by 'realized inorganic CO2 removal' (to be consistent with the recent work of Steinwidder et al. (2025)). In the text (abstract and introduction), we clarify that inorganic CO2 removal is the sum of changes in DIC leaching and soil inorganic C (SIC).

We replaced CDR potential by 'potential inorganic CO2 removal' throughout the manuscript in line with Steinwidder et al (2025). We clarify this term in the abstract, intro and methods. Accordingly, we also modified the title into:

**Weathering without realizing inorganic CO2 removal revealed though base cation monitoring.**

abstract line 24: Here, we investigated realized inorganic  $CO_2$  removal (defined as the sum of the change in dissolved inorganic C leaching and in solid inorganic C).

intro Line 44:. EW relies on accelerating natural weathering reactions of silicate minerals with water ( $H_2O$ ) and carbon dioxide ( $CO_2$ ) (as in **Reactions 1 to 3**), which increases the concentration of base cations and dissolved inorganic C (DIC) in water, delivering inorganic  $CO_2$  removal.

We also adapted the title; earlier we had: "Weathering without inorganic CDR revealed through cation tracing"; The new proposed title is:

Line 34: The potential inorganic CO2 removal, defined as the maximum inorganic CO2 removal achievable if all weathered base cations, adsorbed by soil pools in this experiment, would

leach out of the soil and be fully balanced by carbonate anions, was estimated at 26 kg CO2 t-1 basalt.

**Intro:**

Line 117: From base cations in plant and soil pools, we can thus calculate a 'potential inorganic CO2 removal', a terminology proposed by Steinwidder et al., (2025). This is a maximum quantity of Inorganic CO2 removal that can be achieved when all cations released through silicate weathering are charge-balanced by bicarbonate/carbonates and leached from soils.

**methods:**

Line 250: In addition, we calculate a 'potential inorganic CO2 removal'. We use the same definition for potential inorganic CO2 removal as in Steinwidder et al. (2025). A 'potential inorganic CO2 removal' can be defined as the maximum amount of inorganic CO2 that could be removed if all experimentally determined, weathered, soil-retained base cations were to leach from the soil and be completely balanced by carbonate anions. Potential inorganic CO2 removal was previously 'CDR potential' by Niron et al. (2024)). The concept of CDR potential was first introduced by Phil Renforth (2019) to describe the maximum inorganic CO2 removal achievable if all base cations within a rock were to completely weather. More recently, Beerling et al., (2024) quantified base cation losses from topsoils using an immobile/mobile tracer approach (see Section 4.2), from which they also derived a measure of CDR potential. To maintain conceptual clarity, we avoid using the term CDR potential for purposes other than its original definition by Renforth (2019). When the term is employed, its meaning should always be explicitly stated.

Q5) Title: Avoid using acronyms in titles. Title is not accessible and need to be revised- few will have any idea what "inorganic CDR" is, or "cation tracing"

We agree with the reviewer and have and have revised the title for clarity. We specify that the cations monitored are base cations. We also suggest to use monitoring as a more general term.

We propose the following title as a alternative:

**Weathering without realizing inorganic CO2 removal revealed though base cation monitoring.**

Q6) Abstract: Needs substantial revision to improve clarity and accessibility. The final paragraph especially is vague and qualitative.

We revised clarity of the abstract, taking into account this and other comments by the reviewer, please find the improved abstract below:

**Line 23:**

Enhanced Weathering using basalt rock dust is a scalable carbon dioxide removal (CDR) technique, but quantifying rock weathering and CDR rates poses a critical challenge. Here, we investigated realized inorganic CO2 removal (defined as the sum of the change in dissolved inorganic C leaching and in solid inorganic C) and weathering rates by treating mesocosms planted with maize with basalt (0, 10, 30, 50, 75, 100, 150 and 200 t ha-1) and monitoring them for 101 days. We observed no significant inorganic CO2 removal, as leaching of dissolved inorganic carbon did not increase, and soil carbonate content declined over time.

To gain insights into the weathering processes, we traced the fate of base cations in the soil and plants. This analysis showed that most base cations were retained in the topsoil reducible soil pool, typically associated with iron (hydr)oxides, while increases in the exchangeable pool were about a factor 10 smaller. Soil base cation scavenging exceeded plant scavenging by approximately two orders of magnitude. From the base cations in all pools (soil, soil water and plants), we quantified log weathering rates of -11 mol total alkalinity m-2 basalt s-1. The potential inorganic CO2 removal, defined as the maximum inorganic CO2 removal achievable if all weathered base cations, adsorbed by soil pools in this experiment, would leach out of the soil and be fully balanced by carbonate anions, was estimated at 26 kg CO2 t-1 basalt.

In conclusion, despite clear weathering of basalt rock, we found no inorganic CO2 removal within the timescale of this experiment. The observed increase of aluminum in association with the reducible soil fraction indicate the formation of secondary minerals. These, along

with enhanced base cation exchange capacity, may contribute to long-term soil fertility and promote the stabilization of soil organic matter.

Q7) L24: Do not make subjective comments, e.g., delete "surprisingly and "even" (L25).

we deleted these terms and checked for all other cases where surprisingly and even were used.

Q8) L31: What is CDR potential? Total dissolution of applied rock? Or potential over a time period? The abstract needs to stand alone (many will only read this), so it is essential to use accessible and inclusive language.

See Q4, we clarified this in the abstract, intro and methods.

Q9) L34: refers to time, so "larger" should be replaced with "longer". Please add timeframeshow long do your data suggest? And what timescale is "commonly assumed"? I'm unaware that as yet there are any commonly assumed timescales.

We replaced 'larger' for 'longer'. We deleted the text on commonly assumed timescales and emphasizethe uncertainty of DIC leaching timeframes (see the revised abstract under Q5).

**Introduction:**

Q10) L43: Need to sort out terminology. TA is not a proxy for DIC; these are two distinct variables. TA is not "the sum of base cation charges", linked to this, what is delta in Eq. 1?

We thank the reviewer for this valuable comment. We fully agree that total alkalinity (TA) and dissolved inorganic carbon (DIC) are distinct variables and that TA cannot be considered a direct proxy for DIC. In the revised version, we clarified this conceptual distinction and adjusted our phrasing accordingly. We now explicitly state that TA can be used to estimate DIC through calibration, rather than being a proxy for it.

In addition, we clarified the meaning of  $\Delta$  (delta) as the difference between amended and unamended (control) soils. We moved Equation 1 to the Methods section to improve clarity. We also clarified how  $\Delta TA$  is calculated in our approach, emphasizing that the approximation is based on changes in base cation charges assuming negligible change in conservative anions.

The following changes were made in the revised manuscript:

Line 49: DIC (the sum of aqueous  $[CO_2]$ ,  $[HCO_3^-]$  and  $[CO_3^{2^-}]$ ) can either be measured directly or estimated indirectly from total alkalinity (TA) or electrical conductivity, which are less expensive to monitor and can be empirically linked with DIC through calibration curves (Amann & Hartmann, 2022) (see also **Fig. S10**). This calibration is feasible because, according to the explicit conservative expression for TA in water, TA =  $[HCO_3^-] + [CO_3^{2^-}] + [OH^-] - [H^+]$  (Wolf-Gladrow et al., 2007). TA can also be approximated from the sum of base cation charges, minus the sum of conservative anion charges (e.g. chloride, sulphate, phosphate, nitrate) (Barker, 2013; Wolf-Gladrow et al., 2007).

and

Line 243: We use the delta ( $\Delta$ ) symbol to denote the difference relative to unamended control soil. Accordingly, we quantify  $\Delta TA$  (the change in total alkalinity in the basalt-amended soil relative to the control) based on the difference in base cation concentrations between amended

and unamended soils. As basalt only contains cations and no conservative anions, we assume that  $\Delta TA$  can be quantified from the change in base cation charges (**Equation 1**).

$$\Delta TA \approx 2 * (\Delta Ca + \Delta Mg) + \Delta Na + \Delta K - \Delta conservative anions$$

$$with \Delta conservative anions = 0$$
(1)

Q11) L63: changes in DIC during soil water transport have been known about for years, not recently as indicated here. Be sure to include relevant refs from soils literature.

We agree and revised the text accordingly, including relevant references:

line 76: Quantification of Inorganic CO2 removal by EW has often focussed on tracking DIC or alkalinity leaching in porewaters (Holzer et al., 2023; McDermott et al., 2024). However, it is also important to consider DIC in exported soil water (leachates)(Larkin et al., 2022) as changes in DIC during soil water transport are well-established. Numerous studies demonstrated that soil water movement and pH strongly govern DIC dynamics, both in soil research (Öquist et al., 2009; Schindlbacher et al., 2019) and in EW research (Dietzen et al., 2018; Niron et al., 2024; Reynaert et al., 2023; Vienne et al., 2024).

**Q12) L70: temporally? Is this correct? Or do you mean temporarily?**

We thank the reviewer for catching this language error. The correct term is temporarily, not temporally. We have corrected this in the revised manuscript.

Q13) L90: This paragraph needs revising. DIC and TA are being used interchangeably; the first part refers to DIC then this transforms into "scavenged TA". As above, DIC and TA are not the same thing: make sure they are being used correctly.

We thank the reviewer for this helpful comment. We revised the paragraph to focus exclusively on the exchange between protons and base cations, leading to DIC degassing, and removed references to TA in this context.

To further improve clarity, we also replaced the term "scavenged TA" with "base cation charges retained in the soil" in the text as this more accurately reflects the process we describe and avoids conflating TA with DIC.

Line 112: The undesirable side-effect of base cation scavenging (by plant/soil pools) is the release of protons to maintain charge balance. This release of protons converts negatively charged DIC ( $HCO_3^-$  and carbonate anions ( $CO_3^{2-}$ )) to  $H_2CO_3$ , which is in equilibrium with gaseous  $CO_2$  ( $CO_3^{2-}$  +  $H^+ \rightarrow HCO_3^-$  and  $HCO_3^-$  +  $H^+ \rightarrow H_2CO_3 \Leftrightarrow H_2O + CO_2$  (g)). Hence, inorganic  $CO_2$  removal can be reversed during temporary storage of base cations and realized again when these base cations are released back from soil and plant pools into the aqueous phase.

Line 544: A mass balance of base cations shows that exported TA was negligible compared to scavenged TA the base cation charges that were retained in the soil over the timeframe of our experiment (101 days) (**Table 4**).

Q14) L94: CDR potential is then CDR predicted from measured weathering of base cations from the applied feedstock. It would be helpful to express it as such,

It should be noted also that CDR potential as defined here is not the same as CDR potential defined in Beerling et al. (2024). Making such comparisons is essential as differences in CDR approaches and terminology are leading to *a lot* of confusion even within the EW community, and even more beyond it. and to note also that it is *not* the same thing as CDR estimated using a TiCAT approach, because although TiCAT includes cations held in the exchangeable fraction, it does not include CDR associated with uptake of base cations on carbonates, reducible, oxidisable or secondary mineral pools, since these are retained in the soil.

We refer to our answer in Q4. The definition of CDR potential requires clearer framing and explicit comparison with other approaches used in the EW community.

In the revised manuscript, we now explicitly define realized and potential CDR (cf. Q4) We also clarify how our definition differs from the terminology used in Beerling et al. (2024) and briefly discuss the differences with the Ticat approach in the discussion.

Line 250: In addition, we calculate a 'potential inorganic CO2 removal'. We use the same definition for potential inorganic CO2 removal as in Steinwidder et al. (2025). A 'potential inorganic CO2 removal' can be defined as the maximum amount of inorganic CO2 that could be removed if all experimentally determined, weathered, soil-retained base cations were to leach from the soil and be completely balanced by carbonate anions. Potential inorganic CO2 removal was previously 'CDR potential' by Niron et al. (2024)). The concept of CDR potential was first introduced by Phil Renforth (2019) to describe the maximum inorganic CO2 removal achievable if all base cations within a rock were to completely weather. More recently, Beerling et al., (2024) quantified base cation losses from topsoils using an immobile/mobile tracer approach (see Section 4.2), from which they also derived a measure of CDR potential. To maintain conceptual clarity, we avoid using the term CDR potential for purposes other than its original definition by Renforth (2019). When the term is employed, its meaning should always be explicitly stated.

**Material and Methods:**

Q15) L122 and elsewhere: missing subscripts/superscripts

Adapted and checked throughout the manuscript.

Q16) L227 and elsewhere, make sure all acronyms are spelt out on first use (here, AIC)

OK. We checked whether acronyms were spelt out correctly on first use throughout the manuscript.

Q17) L153 and elsewhere: please report accuracy and precision of all measurements, and how these were determined.

We added these details in a paragraph:

Line 186: Two quality control (QC) standards were analyzed for individual elements (Ca, K, Mg, sodium (Na), silicium (Si) and Fe). The mean precision of the QC standards was 0.84%, 1.12%, 0.54%, 2.79%, 1.67% and 1.30% for the respective elements. The mean accuracy for the two QC standards was 1.87%, 2.30%, 0.17%, 1.88%, 1.39% and 2.65% for Ca, K, Mg, Na, Si and Fe respectively. For TA soil water samples, mean accuracy and precision for two different QC standards were 1.51 and 1.72% respectively. The DIC measurements with FormacsHT had an accuracy and precision 1.09 and 0.23% respectively. Accuracy and precision were determined based on 12 measurements of a QC for TA (standards: 150 and 350 mg CaCO3  $L^{-1}$ ) and DIC (range between 10 and 100 mg  $L^{-1}$ ) and based on eight measurements of two different QC concentrations for each individual element.

Q18) L262 - 270: stating that cations in the feedstock rock were "already weathered initially" is confusing- you just mean that you are correcting for the amount of cations initially present in that fraction in the applied feedstock- so it would be better to say this. Additionally it is not correct to say that these corrections have not been applied before (L266), this was already pointed out by Power et al. (2025).

We thank the reviewer for this clarification. We agree that our original phrasing ("already weathered initially") was confusing. We have revised the text to clearly state that we correct for the cations initially present in each mineral fraction of the applied feedstock, as suggested. We have also corrected our earlier statement and now acknowledge that such corrections have been applied previously by Power et al. (2025).

Line 304: These individual base cations (e.g. Ca in pool j) are calculated from the difference of cations weathered during the weathering operation minus the cations initially present in that fraction of the applied feedstock (Power et al., 2025) (**Equation 7**).

Q19) L274 (& Table 4): how n changes during transport through the soil profile, into rivers & into the ocean is not tested in this study, hence I suggest the endmembers should be n=1 and n=0.5.

Thank you for this valuable suggestion! We adapted the calculations accordingly. The upper n limit was changed to 1 throughout the manuscript and Table 4 was adapted so that n=1.

tine 325: In **Table 4**, we calculated CDR potentials assuming conservative values of  $\eta$ =0.5 (carbonate precipitation scenario) and  $\eta$ =1 (the highest possible  $\eta$  without any downstream DIC losses).

Q20) L294-5: Sentence doesn't make sense. Note also when log SIc>0 minerals have *potential* (*not* "tendency") to precipitate: it is well documented that for example that in rivers, calcite generally does not precipitate until log Sic>1 due to ion inhibition, e.g., by phosphate. L296: delete "perfect"

Thanks for pointing this out. We agree that oversaturation is possible in rivers and added the references of Zhang et al. (2022) which makes this point. We also add the reference of Morse et al. (2007) where carbonate precipitation inhibition by phosphate is discussed.

line 340: Minerals have the potential to precipitate at log SIc >0, although substantial oversaturation of calcite (log Sic > 1) without calcite formation is possible in rivers due to ion

inhibition, e.g. by phosphate (Morse et al., 2007; Zhang et al., 2022). Likewise, minerals are in equilibrium at a log SIc =0 and dissolve if log SIc <0.

Q21) L326: not clear what "gave an even larger (signal)" means- do you mean there was higher accumulation of Mg in the reducible pool compared to the exchangeable? Why is this "borderline significant"- because the variability was very high?

We agree that the phrasing was unclear. What we meant is that Mg accumulation was higher in the reducible pool than in the exchangeable pool, but this trend was only borderline significant because of higher variability at increased basalt amendment rates. We have revised the sentence to clarify this point. :

Line 379: With higher rock amendment, Mg accumulated in the top soil exchangeable pool (p<0.01). The Mg accumulation in the reducible pool was higher compared to the exchangeable pool, but the slope was borderline significant for the reducible pool (p=0.07) due to higher variability in Mg concentrations with increasing basalt amendment.

Q22) L329: You don't know what phase(s) Al is in- you just know that Al is being found *in association* with the oxidisable or reduced fraction, so please change to say this.

We agree that we cannot identify the exact mineral phases of Al, only the operationally defined fractions in which it is associated. We rephrase: Line 500: In addition, the observed increase of aluminum in association with the reducible soil fraction indicate the formation of secondary minerals.

Q23) L334: Delete "even". Why would concs of Na, Fe and Mg decrease in the oxidisable fraction? It is stated elsewhere (e.g., L100) that cations bound in these phases are unlikely to be released. Presumably the TOC content of the 20-30 cm layer did not decrease significantly? To me this hints at artefacts from the leaching procedure.

We removed "even" as suggested.

We agree that elements in the oxidizable fraction are strongly bound and thus unlikely to leach. However, decreases in these elements may occur due to formation of stronger organo-mineral complexes that are not extractable by the Tessier procedure. Although bulk total organic C did indeed not change significantly, shifts between SOC fractions with different binding strengths could still occur (see e.g. Lopez-Sangil & Rovira, 2013 who distinguished seven fractions with increasing binding strength).

To acknowledge this uncertainty, we added a remark in the monitoring section that observed decreases in oxidizable elements may reflect either an artefact of the leaching procedure or a shift toward more stable, unextractable organo-mineral complexes.

Line 610: As discussed in the previous section, another key challenge is that the fate of base cations may remain uncertain if strongly bound crystalline organo-minerals (see Lopez-Sangil & Rovira, 2013) form that are unextractable by the Tessier scheme. Such processes may have contributed to the observed decrease in oxidizable elements at larger depth, although this could also be an artefact of the applied extraction procedure.

Q24) L348: Does this also include Na, K and Ca in the soil pore waters? For the 0-20cm fraction, would be informative to also show the quantity of cations residing in the soil waters. Fig. 6: Some of the text is not legible.

This did not include base cations in the pore water yet. We quantified the minor contribution of bases in the pore water and added it as an additional supplementary Figure (**Fig. S24**).

Compared to the contribution of the soil, contributions of pore water bases were negligible. For Ca and Mg for example, we found an increase in pore waters of 0.76 and 0.33  $\mu$ mol/g basalt (<<< th>than the increase observed in solid soil pools, that were in the 100s of  $\mu$ mol/g basalt for both Ca and Mg).

We refer to Fig. S24 in the caption of Figure 6: line 416: Base cation changes in top soil pore water are not included in this figure as we only include charge equivalent adsorbed by soil pools here, yet base cations in top soil pore water were negligible (see **Fig. S24**).

We adjusted Figure 6 so that annotated p values do not overlap with each other.

**Discussion:**

Q25) L415: delete "shake flask"; L416: delete first "that"

Done. Shake flask was deleted from the text throughout the document and replaced by labscale weathering studies.

Q26) L417: Is the reader expected to know what CDR export via TA was in Amann et al. (2020)? Please make sure all parts of the discussion are clear and transparent.

We thank the reviewer for pointing out the need for greater clarity. We have now included specific values from Amann et al. (2020) and explained how they compare to laboratory weathering rates.

line 475: Absence of substantial DIC leaching is in line with other short-term recent studies (Amann et al., 2020; Larkin et al., 2022; Niron et al., 2024; Vienne et al., 2024). For example, the log Wr of approximately -13 mol TA m2 s-1 quantified from DIC export after 1 year in a mesocosm trial with 220 ton ha-1 olivine-rich rock (Amann et al., 2020) was about three orders of magnitude lower than what would be expected from lab-scale weathering studies (roughly -10 mol TA m2 s-1, (Palandri & Kharaka, 2004)).

Q27) L424: was carbonate present in the initial soil and/or basalt? I.e., was it there to dissolve?

The initial soil was low in SIC, it had a pH of 5.66 and only 0.003% SIC as indicated in Table 1. In the Initial basalt we calculate 0.34% SIC based on the base cations retrieved in the carbonate pool, so yes, carbonates that were already present at the moment of soil amendment may have been dissolved, mainly from the initial basalt.

Q28) L427: Be aware that the cited study neglected to account for application of lime: there is no way that the increased SIC could have come from basalt weathering as it was way too large relative to the basalt application rate.

We assume the reviewer refers to the study of Haque et al. (2020) where wollastonite was applied and the authors indeed acknowledge possible contamination with dolomitic lime. We added a critical note in line 491: A decrease in SIC is in contrast with substantial SIC increases found after wollastonite rock amendment (Haque et al., 2019, 2020). The observed SIC increase in the latter field study may be partly attributed to residual carbonates from prior liming activities instead of new carbonate formation related to silicate weathering (Haque et al., 2020). Thus, not all measured SIC may reflect new carbonate formation in the study of Haque et al. (2020).

**Q29) L433: word missing "exchangeable and pools"?**

Thanks, adapted: line 499: While TA was not exported or taken up by soil carbonates here and plant base cation losses were minor (**Table 4**) it was retained in top soil where the exchangeable and reducible pools reduced solute TA.

**Q30) L435-437: Sentence not clear, seems to be incorrect punctuation and/or words missing**

Rephrased these sentences for clarity: line 510: Our estimate of log Wr, derived solely from significant increases in TA uptake at higher basalt amendment rates, was approximately  $-12 \, \text{mol TA} \, \text{m}^{-2} \, \text{s}^{-1}$ . This estimate reflects changes in the exchangeable and carbonate soil pools, plant uptake, and leachate composition. Notably, this value aligns with previous studies that estimated log Wr values between -12 and -11 based on base cation depletion from the exchangeable pool alone (Kelland et al., 2020; Reershemius et al., 2023; te Pas et al., 2023), as summarized in Vienne et al., (2024).

Q31) L432-443: There have been lots of studies on basalt dissolution, yet only one is briefly mentioned at the end of this paragraph. Expand this discussion to put your data in the context of existing work from the soils literature, which is far more extensive than EW literature.

We agree with the reviewer and have added more studies on the rate of dissolution of basalt from outside the EW literature were added and dissolution of major minerals in basalt (pyroxene and placioclases) were added.

Line 522: From individual application rates, we quantify log Wr ranging between -11 and -10 (Fig. S13); These values are comparable to those observed in soil-free, laboratory-scale basalt dissolution experiments conducted at circumneutral pH (Brantley et al., 2008; Gislason & Oelkers, 2003). They also approximate the dissolution rates of key mineralogical components in basalt (such as plagioclases (between -12 and -9 for Na and Ca endmembers respectively) and augite (-11.97) under room temperature and neutral pH conditions (Gudbrandsson et al., 2011; Hermanska et al., 2022; Palandri & Kharaka, 2004).

Q32) L446 and elsewhere: re-phrase "shake flask" or at the least explain what on earth this means

Shake flask was deleted from the text throughout the document and replaced by lab-scale weathering studies.

Q33) L447: why should base cations be "irreversibly dissolved"? Because there is no uptake on exchange sites and conditions are far from equilibrium? I can believe the latter but it is hard to see how the former can be true.

We agree that "irreversibly dissolved" was not the correct phrasing, and that "far from equilibrium" better describes the intended meaning.

Line 529: Although this and other experiments show relatively consistent weathering rates from exchangeable base cations (comparable to those observed in lab-scale studies) we emphasize that, unlike laboratory conditions where base cations remain far from equilibrium in excess water, soils experience solid-phase cation scavenging, which promotes DIC degassing (**Figure 1**).

Q34) L449: delete "only"- again, subjective comments have no place in scientific writing!

Thank you, we checked for subjective comments such as surprisingly and only followed by a value throughout the manuscript.

Q35) L457: add "over the timeframe of our experiments (111 days)" to the end of this sentence

OK. Line 544: Furthermore, for climate change mitigation, not only the amount of potential inorganic CO2 removal is important, but also the timescale at which this CDR is realized (Kanzaki et al., 2025). A mass balance of base cations indicates that exported TA was negligible compared to base cation charges that were retained in the soil over the timeframe of our experiment (101 days) (**Table 4**).

**Q36) L465: I think also Li isotopes, see Pogge von Strandmann papers**

Yes, we added his paper from 2022. Line 553: Clay formation has previously been suggested for EW application based on changes in soil water Ge/Si ratios and Si isotopes (Vienne et al., 2024) and also based on Li isotope measurements (Pogge von Strandmann et al., 2022).

Q37) L467: change to "such as kaolinite that do not sequester base cations".

Adapted.

Q38) L469: let's hope EW doesn't lead to chrysotile formation! The implications of this for human health would swamp the CDR effect......

We also hope that this does not happen. In our group, we also do dunite reactor experiments, preliminary XRD data of fines formed in the reactor suggest that mainly chlorite and smectite group minerals were formed for dunite in our experiment.

Q39) L473-480: see comment above. I don't think this paragraph is helpful, given that clays are not expected to be in the reducible fraction.

We agree with the reviewer and refer to Q1 and Q3. We added that we expect (hydr)oxides in the reducible fraction, with a critical note and mention that some crystalline clays can be extracted as well by the chemicals that were added with the intention to extract hydroxides and organic matter (see the work of Ryan et al. (2008)).

Line 499: While TA was not exported or taken up by soil carbonates here and plant base cation losses were minor (**Table 4**) it was retained in top soil where the exchangeable and reducible pools reduced solute TA. We expect cations to be primarily associated with Fe- and Mn-(oxyhydr)oxides in the reducible pool and with organic matter in the oxidizable pool, as supported by literature (Tessier et al., 1979); However, the extraction chemicals of this sequential extraction scheme (hydroxylamine and H2O2) are known to have a limited specificity and may have also partially targeted other mineral phases (such as clays) (Ryan et al., 2008), which could explain the elevated Si observed in the topsoil pools (**Fig. S15**). In addition, the observed increase of aluminum in association with the reducible soil fraction indicate the formation of secondary minerals. While we cannot pinpoint exactly what Mg-phases were formed in our soils, our results do demonstrate substantial base cation retention in the soil and show that there can be more base cation losses to soils than to the exchangeable pool alone.

**Q40) L483: what are "the latter clays"? Not clear to me.**

This referred to base cation bearing clays such as smectites or illite for example. This is clarified in the revised manuscript,

on line 564: Although unfavourable for inorganic CO2 removal, if base cation bearing secondary clay minerals would form, they can increase SOC (Georgiou et al., 2022; Heckman et al., 2022; Steinwidder, Boito, Frings, Niron, Rijnders, De Schutter, et al., 2025). Georgiou et al. (2022) refer to base-cation bearing clays (e.g. smectitic or illitic clays) as 'high-activity minerals' due to their higher SOM stabilization capacity compared to secondary minerals that do not contain base cations (i.e., 'low-activity minerals', with a lower CEC such as kaolinite).

Q41) L491: Delete last sentence of this paragraph.

We agree that the sentence was redundant and deleted it.

Q42) L503: Avoid referencing non-peer reviewed articles. Here, would be better to quote e.g., Clarkson et al. 2024

Changed the reference of isometric to Clarkson et al. 2024, thanks.

Q43) L511: I think erosion is accounted for, that is the reason for normalising to Ti?

Excellent point. After discussion this again with Tom Reershemius at the ERW 25 conference we realized that the original wording was not fully correct. For feedstocks that have Ti, the TiCat approach indeed accounts for physical transport by erosion or bioturbation. We have revised the paragraph accordingly to clarify this point and to include the limitations of the approach.

Adapted in line 583: A first soil measurement approach is a total cation accounting approach, which quantifies the loss of base cations from top soils (te Pas et al., 2025). However, this approach only focuses on the top soil and fails to account for physical cation transport from top soils due to erosion or vertical feedstock transport via infiltration or bioturbation. Alternatively, in a mobile/immobile tracer element approach (often named 'TiCat' by the EW community), cation losses from amended top soils are quantified along with immobile tracers, which can account for cation losses through bioturbation or erosion (Reershemius et al., 2023). Nonetheless the disadvantage of TiCat is that it does not track potential TA scavenging (e.g. by organic matter or clays) at larger depth. Our potential inorganic CO2 removal estimate will thus differ from a potential inorganic CO2 removal estimate quantified using a TiCat approach.

**Conclusions:**

Q44) L545: Change to: "....may not immediately lead to inorganic CDR benefits."

Adapted, line 626: Our findings indicate that basalt-based enhanced weathering may not immediately lead to the inorganic CO2 removal previously anticipated in projections and IPCC reports (Babiker et al., 2022; Minx et al., 2018).

Q45) L548: change "proving" to "demonstrating"

**Adapted.**

Q46) L549: "reducible bases"???? assuming the bases are Ca, Mg, Na and K these are not reduced in terrestrial environments. Correct the phrasing.

Adapted, line 630: Additionally, we observed a borderline significant but substantial increase in base cations in the reducible topsoil pool with greater basalt application, which may further suppress TA leaching.

Q47) L551: Well maybe but in practise basalt application rates will be at the lower end of your experiments, i.e. <30 t/ha. Please link this to reality.

We agree that the basalt application rates used in real-world applications may be <30 t ha-1, which is more practical to apply. Moreover, the effectiveness of EW may be higher due to lower base cation exchange.

Adapted line 632: As base cation exchange increased with higher basalt amendments, we infer that greater application rates can further delay the release of DIC from soil minerals to surface waters. However, in practice, EW is typically applied at application rates below 30 t ha-1. These lower application rates, which are more practical to apply, may enhance the effectiveness of inorganic CO2 removal by reducing lag times for DIC release.

**Suppl info:**

Q48) I assume all data will be supplied?

Yes, I added the zenodo link with all data to the reuploaded pdf. We added a data and code availability statement:

Data and code used in this manuscript are freely available at: https://zenodo.org/records/15129984

Q49) Fig. S6, S11 etc: Wherever standard error is reported, please give n

We added N for every applicable figure, thanks.

---

## Author Comment (AC3)

Reviewer 2: In CDR, the idea is that alkaline rock captures CO2, while generated secondary minerals and increased cation exchange capacity (CEC) could enhance plant productivity and soil organic carbon (SOC) retention in soils. By studying different soil fractions, authors show that CO2 capture would be minimal, thus MS is tuning down hopes that basic rocks can mitigate CO2 buildup to atmosphere. Manuscript is interesting to me, who is novice about the subject. MS gives an idea how difficult it is to estimate how much carbon is ultimately stored at least for longer periods and how to estimate it.

We thank reviewer 2 for the interest in our findings.

Q50) MS gives a good introduction to subject, even could be in many points more accurately written (lot of typos) and also quite lengthy (25 pages on supplementary material). My comments may be inaccurate and not totally on topic. Furthermore, I cannot comment much on methods. I am wondering about some conventions on this branch of soil science, and also supriced how far "real" carbon balance measurement community is from CDR world. This said as an excuse to begin with.

We sincerely thank the reviewer for their thoughtful comments and for openly stating the limits of their expertise in some areas. We appreciate the time and care taken to engage with the manuscript.

Regarding the length of the supplementary material, we agree that it is extensive. However, due to the large number of analyses and visualizations required to transparently document our results, there was no suitable alternative to placing them in the supplement. We felt this was the most appropriate way to ensure full reproducibility and clarity without overwhelming the main manuscript.

Q51) If I got it right, the benefit for long term CDR "0.4-0.5 kg CO2 ton-1 basalt or 0.020-0.025 tCO2 ha-1 for a basalt application rate of 50 t ha-1CDR" by enhanced weathering. So total climatic impact is rather the opposite to aims of basalt rock dust application. When energy consumption on transport, spreading and crushing of basalt is added to equation. However, this is not said enough clearly, at least to a reader which is not an expert on this subject.

We thank the reviewer for this important observation. We agree that the net climate impact of enhanced weathering must account not only for  $\mathrm{CO}_2$  removal through basalt application but also for emissions associated with the sourcing, crushing, transport, and spreading of the material. In the revised manuscript, we now make this point explicit in the discussion. We also note that the updated long-term CDR potential, after increasing  $\eta$  to 1 (cf Q19) and including non-significant alkalinity slopes, was 26.33 ± 6.13 kg  $\mathrm{CO}_2$  ton-1 basalt (Table 4).

For comparison, Lefebvre et al. (2019) estimated that, in Brazil, for 65 km transport, 75 kg CO2 is emitted/ton CO2 removed by enhanced weathering due to life cycle emissions. They assume 0.225 ton CDR/ton basalt: 75 kg CO2 emission / t CDR \* 0.225 ton CDR/ ton basalt = 16.87 kg CO2 emitted/ton basalt applied. This is substantial and should be included in every C crediting methodology.

We incorporated this in the discussion, on line 540: Moreover, life-cycle emissions associated with mining, grinding and transporting rock are typically of the same order of magnitude as our relatively low potential inorganic CO2 removal (Lefebvre et al., 2019).

Q52) Reference to earlier studies near conclusions showing benefits of CDR, would help to understand this statement: "Our results suggest that EW using basalt amendments may not yield the immediate inorganic carbon dioxide removal (CDR) benefits previously anticipated".

We agree with the reviewer and have clarified that the "previously anticipated" CDR benefits refer to projections from earlier studies and assessments, such as those discussed in Minx et al. (2018), which highlighted enhanced weathering as a promising negative emissions technology. We have therefore added this reference to provide clearer context. In addition, we add chapter 12 of the IPCC report from 2022 (Babiker et al. 2022).

line 626: Our findings indicate that basalt-based enhanced weathering may not immediately lead to the inorganic CO2 removal previously anticipated in projections and IPCC reports (Babiker et al., 2022; Minx et al., 2018).

Q53) As I have been measuring mostly  $CO_2$  as flux out from soil and DIC in lake water columns, I wonder how big emphasis is put on DIC concentration in MS to soil water DIC. And also to DIC flooding to sea. (When nothing got trough experimental mesocosmoses). Sea as the fate of carbon and the base level where to compare everything related to greenhouse gases? Why not Global Warming potentials used by IPCC? Maybe explained more in introduction, why conventional flux measurements are not enough, may help people like me to understand better why this methodology is used, As a novice, I think that just measuring CO2 emission with closed chamber in dark, measuring carbon gain to plant biomass (done here?) would add quite much to carbon capture (or carbon dioxide capture) measurements, when also DOC concentration is measured (done).

We thank the reviewer for this thoughtful comment. We agree that We agree that a comprehensive greenhouse gas (GHG) budget would ideally include all carbon fluxes and pools, including  $CO_2$  fluxes from root respiration, SOM respiration, DOC, DIC, and biomass changes.. However, the focus of our study follows current EW monitoring and certification frameworks, which focus primarily on inorganic C removal via DIC export to the ocean, as this is considered the most durable form of C sequestration. We have clarified this rationale in the Introduction to help non-specialist readers understand why we focus on DIC and base cation dynamics rather than a full carbon balance.

Line 46: In this study, we focus on DIC export from soils to the ocean, as this pathway is considered the most durable form of carbon sequestration (Renforth & Henderson, 2017), rather than aiming to quantify a full greenhouse gas budget.

We also refer to complementary studies providing additional insights into plant biomass changes in this experiment (Rijnders et al., 2025). For an in-depth analysis of changes in CO2 fluxes, SOM and DOC we refer to the work of Boito et al. (2025 and Steinwidder et al. (2025).

In the revised manuscript we included this information on line 138: This experiment was part of a larger mesocosm experiment that aimed to investigate heavy metal fate and plant biomass in silicate amended maize plants (Rijnders et al., 2025).

And line 564: Although unfavourable for inorganic CDR, if base cation bearing secondary clay minerals would form, they can increase SOC (Georgiou et al., 2022; Heckman et al., 2022; Steinwidder, Boito, Frings, Niron, Rijnders, De Schutter, et al., 2025).

Q54) And still how to separate DIC coming from root exudates from that coming from weathering (without using carbon stable isotopes to label plant root exudates).

Thank you for this question. Indeed roots can also exude HCO3-, it is unclear if the amount of exudation of HCO3- by roots would be affected by basalt amendment. Nonetheless, we quantifiedy the weathering rate based on cation tracing, not on DIC. We added a sentence in the discussion to clarify that we cannot distinguish between DIC coming from root exudates and DIC derived from rock dissolution.

Line 470: Increased DIC in basalt soils relative to controls may result from enhanced plant root respiration or DIC exudation or from mineral weathering; our dataset does not allow these effects to be separated.

Q55) Authors note that "**The undesirable side-effect of base cation scavenging** (by plant/soil pools) **is release of CO2**". I totally agree to this as a technical problem for measurements, but not as one who has done some respiration measurements and know that microbes do a lot, pH has an effect, out salting due to fertilization, concentration difference between atmosphere CO2 and DIC, wind and temperature also affect CO2 fluxes.

We think this was a misunderstanding. While we fully agree with the reviewer the soil CO2 efflux also depends on temperature, moisture, pH, here we were referring to degassing of inorganic C.

We rephrased this sentence to avoid confusion:

Line 112: The undesirable side-effect of base cation scavenging (by plant/soil pools) is release of protons through charge balance, which convert negatively charged DIC ( $HCO_3^-$  and carbonate anions ( $CO_3^{2-}$ )) to  $H_2CO_3$  which is in equilibrium with gaseous  $CO_2$  ( $CO_3^{2-} + H^+ \rightarrow HCO_3^-$  and  $HCO_3^- + H^+ \rightarrow H_2CO_3 \Leftrightarrow H_2O + CO_2(g)$ ).

Q56) I think text needs to be clarified for broader audience, terms like CDR, TA, DIC, MRV approach etc. explained more clearly.

We agree with the reviewer and have more clearly defined CDR, DIC and TA. MRV was removed from the text:

CDR: As also addressed in Q4, we use (potential / realized) inorganic CO2 removal and avoid use of 'inorganic CDR'. We do keep the abbreviation CDR as this is a commonly used acronym. We state the importance for CDR in the first line of the intro:

Line 42: To meet the "well below 2°C warming" target established by the United Nations' Paris Agreement, Carbon Dioxide Removal (CDR) must complement conventional climate change mitigation efforts (Minx et al., 2018).

We added a clear definition for TA (total alkalinity), see Q10.

DIC was better introduced: Line 49: DIC (the sum of aqueous  $[CO_2]$ ,  $[HCO_3^-]$  and  $[CO_3^2^-]$ ) can either be measured directly or estimated indirectly from total alkalinity (TA) or electrical conductivity, which are less expensive to monitor and can be empirically linked with DIC through calibration curves (Amann & Hartmann, 2022) (see also **Fig. S10**).

MRV: we replaced this by "monitoring C sequestration" to make it clearer to a broader audience, we replaced MRV by monitoring where needed, for example in these sentences:

MRV deletion: Line 131: Here, we make a mass balance after 101 days of experiment, investigate the fate of base cations through exploration of sequential extractions as <del>an MRV</del> a monitoring strategy for weathering and implications for C sequestration.

Line 602: However, this MRV monitoring approach involves complexities such as feedstock correction, leaching solution strength and soil heterogeneity.

Q57) Not start sentences with chemical abbreviations, like "Al" instead of Aluminum etc.

We screened the entire document and defined atoms/molecules when first used.

Q58) There were also lot of typos, mistakes and even repetition in reference list and also missing journal names. Easily corrected things, but these are annoying to reader. Even my comment are critical, still this MS has potential to be published in SOIL journal.

We thank the reviewer for noting these issues. We have carefully rechecked the entire reference list and corrected all typographical errors, missing journal names, and duplicated entries. Specific corrections include:

- -Preprint that did not have a journal Kanzaki et al. (2024) was published after submission, replaced by Kanzaki et al. (2025)
- -The preprints of Reershemius et al. (2023) and Reynaert et al. (2023) were cited in stead of the published MSs, these are now replaced by the correct citations of the published manuscript.
- -Isometric, 2024 (not a scientific paper, replaced by Clarkson et al., 2024) see also Q42.
- -Palandri et al. (2004) is the only reference still cited without journal name as this is a report from the USGS and not a scientific article

Q59) Some general notes, only few of the typos are shown in list. the name of MS ... inorganic CDR..., which is not written open thus hard to get interested at rapid look to those not familiar with CDR acronym.

We adapted this and refer to the modified title in Q5. In the revised manuscript, we consistently use inorganic CO2 removal.

Q60) "SOM bound to cations in the exchangeable pool is expected to be more prone to microbial decomposition than SOM bound in the oxidizable pool", a reference here after, or was it earlier?

We agree that this statement should be supported by a reference. We have added a citation to Poeplau et al. (2018) and clarified the reasoning by specifying that SOM in the exchangeable pool is extracted with weak salt solutions and typically represents the more labile carbon fraction with faster turnover.

line 206: SOM bound to cations, extracted with weak salt solutions in the exchangeable pool typically has a low turnover time (Poeplau et al., 2018) and is therefore thought to be more susceptible to microbial decomposition than oxidizable SOM.

Q61) 208 "in all aboveground biomass parts: stems, leaves, flowers and corn ears". is it corn or maize, and ears?

Thanks for noticing that there was still a mention of 'corn' we decided to use maize throughout rather than corn. I fixed two cases where corn was still mentioned.

An "ear of maize" is the harvested part of a maize plant that contains the kernels, a cluster of seeds arranged on a cob and protected by a husk. We previously used corn fruit, but were advised that the correct terminology is a maize ear.

**Supplement materials**

Q62) Conductivity and DIC correlation jump in at Suplemental material, not discussed earlier. Is it a autocorrelation with amount of fertilizers etc.. maybe more discussion on this on main article, not discussion in supplement?

We agree that the relationship between DIC and electrical conductivity should be introduced in the main text rather than appearing only in the Supplement. We have now linked Figure S10 to the corresponding discussion in the main manuscript and clarified that these relationships are used to derive DIC from more easily measurable parameters, as described by Amann & Hartmann (2022).

Line 49: DIC (the sum of aqueous [CO2], [HCO3-] and [CO32-]) can either be measured directly or estimated indirectly from total alkalinity (TA) or electrical conductivity, which are less expensive to monitor and can be empirically linked with DIC through calibration curves (Amann & Hartmann, 2022) (see also **Fig. S10**).

Q63) In supplementary figure, Ph, DIC, TA and DOC conc. increase at 100 and 150 tn ha-1 application. pH may be the driving force, so scale better so that 7.0 is visible (now 6.8 - 7.2). (I think it is the border between acid and base - in water)

We adapted Fig. S6, so that the y axis of the pH panel (Fig. S6A) has axis ticks at every 0.2 units of pH rather than at every 0.4, so that it shows exactly where pH 7 is.

Q64) In sup. Fig S8 exp. days and application rate or captions are mixed, there was not 200 days in experimnt.

This has been corrected.

Q65) Unclear in table S4: Effects and significance: is this time or application amount in regression?

The effect of application amount in column 2 and time x application amount interactions in column 3, we indicated this in the first row of the table. N.S. = not significant.

| Top soil pore water (0-10 cm) |               |               |
|-------------------------------|---------------|---------------|
| Parameter                     | Basalt effect | Time x basalt |
| (unit)                        | and p-value   | interaction   |
|                               |               | effect        |
|                               |               | and p-value   |
| pH (-)                        | +6.9 e-3      | N.S.          |
|                               | (p<0.01)      |               |
| DIC (mg/L)                    | +1.6 e-2      | +4.9 e-4      |
|                               | (p=0.27)      | (p=0.04)      |